# Pruning Neural Networks via Coresets and Convex Geometry: Towards No Assumptions

**Murad Tukan**[*][†]
muradtuk@gmail.com

**Loay Mualem**[*][†]
loaymua@gmail.com

**Alaa Maalouf**[*][†]
alaamaalouf12@gmail.com

## Abstract

Pruning is one of the predominant approaches for compressing deep neural networks (DNNs). Lately, coresets (provable data summarizations) were leveraged for pruning DNNs, adding the advantage of theoretical guarantees on the trade-off between the compression rate and the approximation error. However, coresets in this domain were either data-dependent or generated under restrictive assumptions on both the model's weights and inputs. In real-world scenarios, such assumptions are rarely satisfied, limiting the applicability of coresets. To this end, we suggest a novel and robust framework for computing such coresets under mild assumptions on the model's weights and without any assumption on the training data. The idea is to compute the importance of each neuron in each layer with respect to the output of the following layer. This is achieved by a combination of Löwner ellipsoid and Caratheodory theorem. Our method is simultaneously data-independent, applicable to various networks and datasets (due to the simplified assumptions), and theoretically supported. Experimental results show that our method outperforms existing coreset based neural pruning approaches across a wide range of networks and datasets. For example, our method achieved a $62\%$ compression rate on ResNet50 on ImageNet with $1.09\%$ drop in accuracy.

## 1 Introduction and Backround

Deep neural networks (DNNs) achieved state-of-the-art (SOTA) performance on a large variety of tasks, e.g., in computer vision [33, 50] and natural language processing (NLP; [87, 20]). However, DNNs usually contain millions or even billions of parameters in order to achieve SOTA performances resulting in large storage requirements and long inference time. This is obstructive when, e.g., dealing with limited hardware or real-time systems such as autonomous cars and text/speech translation. To this end, a large body of research is dedicated to reducing the size and inference costs of DNNs.

**Pruning.** A dominant approach widely used for reducing the size of DNNs is to utilize a pruning algorithm to remove redundant parameters from the original, over-parameterized network. In general, pruning can be categorized into two main types: (i) Unstructured pruning [31, 6] reduces the number of non-zero parameters by inducing sparsity into weight parameters, which can achieve high compression rates but requires specialized software and/or hardware in order to achieve faster inference times. (ii) Structured pruning [35, 60, 77] modifies the structure of the underlying weight tensors, by removing filters/neurons from each layer, usually resulting in smaller compression rates while directly achieving faster inference times with no specialized software; see section 4

---

[*]These authors equally contributed to this paper.
[†]Department of Computer Science, University of Haifa

36th Conference on Neural Information Processing Systems (NeurIPS 2022).

## 1.1 Coresets for Pruning

Notably, many recent papers focused on various types of filter pruning [88, 79] potentially due to the empirical observation that existing filter pruning approaches consistently yield impressive results. However, most pruning methods are based on heuristics, lacking theoretical guarantees on the trade-off between the compression rate and the approximation error. This was the motive for introducing coresets [82, 60] to the world of pruning.

**Coresets.** In machine learning, we are (usually) given an input set $P \subseteq \mathbb{R}^d$ of $n$ points, its corresponding weights function $w : P \to \mathbb{R}$, a feasible set of queries $X$, and a loss function $\phi : P \times X \to [0, \infty)$. The tuple $(P, w, X, \phi)$ is called *query space*, and it defines the optimization problem at hand. For a given problem that is defined by its query space $(P, w, X, \phi)$, and an error parameter $\varepsilon \in (0, 1)$, an $\varepsilon$-coreset is is a small weighted subset of the input points that approximates the loss of the input set $P$ for every feasible query $x$, up to a provable bound of $1 + \varepsilon$.

Since coresets approximate the cost of every query, traditional (possibly inefficient) algorithms/solvers can be applied on coresets to obtain an approximation of the optimal solution on the full data, using less time and memory; see Section B.2 in the appendix for more details.

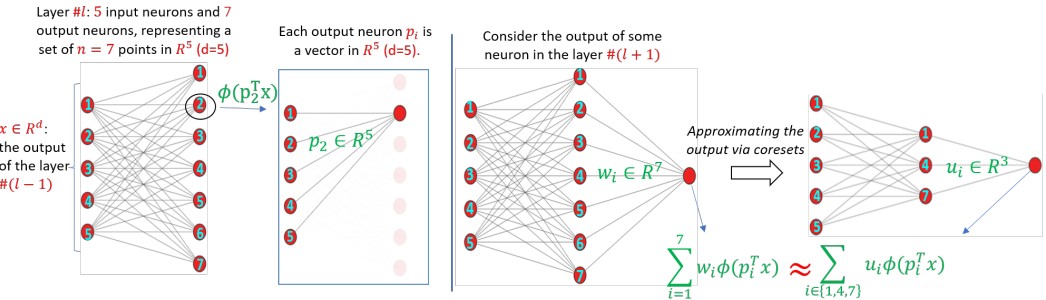

Figure 1: Illustration of our neuron coreset construction on a toy example.

**Pruning via coresets.** Recently, some inspiring innovative frameworks [82, 60, 5, 7] leveraged the idea of coresets for pruning DNNs. Any layer $\ell$ can be represented as a set $P = \{p_1, \cdots, p_n\}$ of $n > 1$ points in $\mathbb{R}^d$, where $d$ is the number of input neurons, and $n$ in the number of output neurons, i.e., each point $p \in P$, represents a specific neuron using its $d$ weights (parameters). When the layer receives an input vector $x \in \mathbb{R}^d$, it outputs the vector $(\phi(p_1^T x), \cdots, \phi(p_n^T x))$, where $\phi : \mathbb{R} \to \mathbb{R}$ is an activation function which defines a non-linear mapping. Focusing on a single neuron $\eta$ in the layer that follows $\ell$, defined by its corresponding vector of weights $w = (w_1, \cdots, w_n)$, we set in the context of coresets, $w(p_i) := w_i$ for every $i \in \{1, \cdots, n\}$ - this is just a mapping from $p_i$ to $w_i$ to simplify the writing and reading. Note that the output of this neuron is $\sum_{p \in P} w(p)\phi(p^T x)$. Assuming that we are given an $\varepsilon$-coreset $(C, u)$ for the query space $(P, w, \mathbb{R}^d, \phi)$ where $C \subseteq P$, and $u : C \to \mathbb{R}$, we have that $(C, u)$ approximates the output of this specific neuron $\eta$ for every query using less parameters; see Figure 1. To formalize the stated above, we now define coresets in the context of activation functions.

**Definition 1.1** (Coreset for activation functions). Let $\varepsilon \in (0, 1)$, and let $(P, w, \mathbb{R}^d, \phi)$ be a query space. Then the pair $(C, u)$, is an $\varepsilon$-coreset for $(P, w, \mathbb{R}^d, \phi)$ if (i) $C \subseteq P$, (ii) $u : C \to [0, \infty)$, and (iii) for every $x \in X$, $\left| 1 - \frac{\sum_{q \in C} u(q)\phi(q^T x)}{\sum_{p \in P} w(p)\phi(p^T x)} \right| \leq \varepsilon$.

Since $C$ is a subset of $P$, we can remove (assign zero to) all weights from $w$ that corresponds to points not chosen to be in $C$ from $P$, and replace the weights of the chosen points in $C$ with the new weights vector $u$; see Figure 1 in [82] for a visual illustration. **To prune neurons,** we refer the reader to Section 2.5 as it is a simple extension. Prior work showed that such approaches successfully result in high compression rates across a wide range of networks and datasets, and even achieves SOTA performance on a verity of them.

The main (strong) advantage of the coreset approach over others was the provided provable theoretical guarantees on the tradeoff between the compression rate and the approximation error, which supports

worse case scenarios. In addition, coresets play an important role in improving the generalization properties of the trained networks [5, 78].

**Sensitivity sampling for constructing pruning coresets.** To compute such coresets, both [60, 82] utilised the known sensitivity sampling framework [9, 52]. In short the sensitivity of a point $p \in P$ in some query space $(P, w, X, \phi)$ corresponds to the importance of this point with respect to the query space at hand, and it is defined as $s(p) = \sup_{x \in X} \frac{w(p)\phi(p,x)}{\sum_{q \in P} w(q)\phi(q,x)}$ - where the denominator is not equal to zero. Once we bound these sensitivities, we can sample points (neurons) from $P$ according to sensitivity bounds, and re-weight the sampled points to obtain a coreset. The size of the sample is proportional to the sum of these bounds. See Section B.1 and Theorem B.2 for more details in the Appendix.

## 1.2 Our contribution

Prior coreset methods for pruning DNNs either (i) imposed restrictive assumptions both on the model's weights and inputs [82], i.e., the input set $P$ representing the neurons, and the query set $X$ which represents the inputs of the layer, are enclosed in a ball in $\mathbb{R}^d$ of radius $r_1$ and $r_2$, respectively, or (ii) the methods are data-dependent, i.e., use a mini-batch of the input set to measure the influence of each parameter on the loss function [5, 60].

To this end, in this work, we take coresets a step further into the realm of pruning by introducing a unified framework with provable guarantees for pruning DNNs (weights and neurons/filters) while minimally affecting the generalization error. Our main improvement is that our framework is simultaneously (i) **data-independent**, (ii) **requires a single assumption** on the model's weights, and (iii) **provably guarantees a multiplicative factor approximation**, which is favourable upon additive approximations; see Theorem 2.6. The approach is based on the widely used theory of coresets allowing us to suggest a provable guarantee on the tradeoff between the approximation error and compression rate for each layer.

We conducted experimental results which established new SOTA benchmarks for structured pruning via coresets across a wide range of networks and datasets. We share all of our resulted models [14].

## 2 Method

In general, the coreset (for pruning) technique hinges upon the insight that any linear layer such as convolutions, can be casted as a matrix multiplication [82]. Hence, we focus in what follows on fully connected (FC) layers, while the details holds for any linear layer. Furthermore, for simplicity, we assume in what follows that the weights of $P$ are all equal to 1 and thus our query space is denote by $(P, \mathbb{R}^d, \phi)$, and the sensitivity of a point $p \in P$ is simply $s(p) = \sup_{x \in X} \frac{\phi(p,x)}{\sum_{q \in P} \phi(q,x)}$. Note that our proofs are easily extended to the general case where we are given a weight function $w : P \to \mathbb{R}$ as discussed in Section 2.5.

### 2.1 Preliminaries

**Notations.** For a positive integer $n$, we use $[n]$ to denote the set $\{1, \ldots, n\}$. For $c \in \mathbb{R}^d$ and a symmetric positive definite matrix $G \in \mathbb{R}^{d \times d}$, we define $E(G, c) := \left\{ x \in \mathbb{R}^d \middle| (x - c)^T G (x - c) \le 1 \right\}$ to be the ellipsoid defined by $c$ and $G$. For an ellipsoid $E(G, c)$, each endpoint of a semi principal axis is called a vertex of $E(G, c)$. We define $\mathrm{rank}(P)$ for any set $P \subseteq \mathbb{R}^d$ to be the dimension of the affine subspace that $P$ lies on. For a set $P \subset \mathbb{R}^d$ the convex hull of $P$ is denoted by $\mathrm{Conv}(P)$. Finally, vectors are treated as column vectors.

### 2.2 Novelty - Löwner ellipsoid meets Carathéodory

Our method hinges upon a combination of two known tools from convex geometry. The novelty of our approach exploits the following observation. Most activation functions $\phi$ are continuous non-decreasing functions, which indicate that for every query $x$ and a set of points $P$, the maximal contribution to $\sum_{p \in P} \phi(p^T x)$ with respect to such activation function is associated to a point on the convex hull of $P$. By finding a geometrical body $B$ of bounded number of vertices, that is (i)

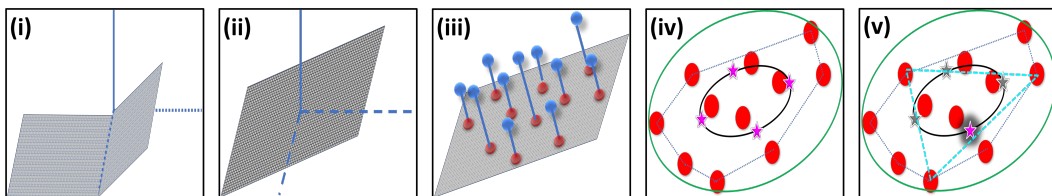

Figure 2: Our novelty in a nut-shell: The very first steps of our technique rely on bounding the ReLu activation function by the $\ell_1$-regression loss function, e.g., for ReLU $\left(p^T x\right)$, where $p = (1, 0)$ in this example (shown in (i)), we first bound it by the $\ell_1$-regression loss function (shown in (ii)) using Definition 2.3. Following this step, a set of points $P$ can be projected into a low dimensional subspace of dimension rank $(P)$ using any dimensionality reduction algorithm as presented in (iii), resulting in the set $P'$. (iv) The convex hull (blue dashed lines) $P'$ (red points) is enclosed by its Löwner ellipsoid (depicted in green). (v) Finally, for each vertex (magenta star) of the enclosed ellipsoid (black ellipsoid), its Carathéodory set is found (red points connected by cyan dashed lines).

enclosed in Conv $(P)$ and (ii) with some dilation factor (expanding) $B$ enclose Conv $(P)$, we will be able to represent each point $p$ on the boundary of the convex hull of $P$ as a convex combination of two points $p_1, p_2$, one of each $(p_1)$ on $B$ and the second $(p_2)$ on its dilated form, which is formalized as the set $\{\alpha(x - c) + c \mid x \in B, \forall \alpha \in [0, 1]\}$, where $c$ here denotes the center of $B$. For such task, Löwner ellipsoid is leveraged.

**Theorem 2.1** (John-Löwner ellipsoid[42]). *Let $L \subseteq \mathbb{R}^d$ be a set of points such that the convex hull of $L$ has a nonempty interior. Then, there exists an ellipsoid $E(G, c)$ (also known as the MVEE), where $G \in \mathbb{R}^{d \times d}$ is a positive definite matrix and $c \in \mathbb{R}^d$, of minimal volume such that $\frac{1}{d}(E(G, c) - c) + c \subseteq \text{Conv}(L) \subseteq E(G, c)$. If $L$ is symmetric around the center $c$, then the dilation factor can be reduced to $\frac{1}{\sqrt{d}}$.*

---

**Algorithm 1:** $\ell_\infty$-CORESET $(P, m)$

**Input** : $P \subseteq \mathbb{R}^d$ of $n$ points with rank $r$
**Output** : A subset $S \subseteq P$ that satisfies
         Lemma 2.5

1 $(Y, z) :=$ affine subspace that $P$ lies on,
  i.e. $P \subseteq \left\{ xYY^T + z \middle| x \in \mathbb{R}^d \right\}$

2 $P' := \left\{ p' := pY^T \right\}$

3 Let map $: P' \to P$ a function that maps
  every $p' \in P'$ to its corresponding point
  $p \in P$

4 $S := \emptyset; K := \emptyset$

5 $E(G, c) := \text{MVEE}(\text{Conv}(P'))$

6 $V :=$ the vertices of ellipsoid
  $\frac{1}{r}(E(G, c) - c) + c$

7 **for** *every $v \in V$* **do**

8    $K :=$
    $K \cup \text{CARATHEODORY-SET}(v, P')$
    `// See Algorithm 4 in the`
    `supplementary material.`

9 **end**

10 $S := \{\text{map}(q) | q \in K\}$

11 **return** $S$

---

**Algorithm 2:** CORESET $(P, m)$

**input** : A set $P \subseteq \mathbb{R}^d$ of $n$ points, and a
      sample size $m$
**output** : A weighted set $(C, u)$

1 $Q := P, i := 1, C := \emptyset$

2 **while** $|Q| \geq 2\text{rank}(Q)^2$ **do**

3    $S_i := \ell_\infty$-CORESET $(Q)$

4    **for** *every $p \in S_i$* **do**

5      $s(p) := \frac{2\text{rank}(Q)^{1.5}}{i}$

6    **end**

7    $Q := Q \setminus S_i, i := i + 1$

8 **end**

9 **for** *every $p \in Q$* **do**

10    $s(p) := \frac{2\text{rank}(Q)^{1.5}}{i}$

11 **end**

12 $t := \sum_{p \in P} s(p)$

13 $C :=$ an i.i.d sample of $m$ points from $P$,
  where each $p \in P$ is sampled with
  probability $\frac{s(p)}{t}$.

14 $u(p) := \frac{t}{m \cdot s(p)}$ for every $p \in C$

15 **return** $(C, u)$

---

Afterwards, $p_1$ and $p_2$ should be represented by points from $P$. Each point on $B$ (specifically, $p_1$) can be represented via a convex combination of $d + 1$ points from $P$. The same holds for points on the dilated form of $B$ (e.g., $p_2$) but via a conical combination (linear combination where the weights are non-negative and the sum of weights is not necessarily 1). This problem is solved by invoking Carathéodory theorem.

**Theorem 2.2** ([10, 95]). *For any $A \subset \mathbb{R}^d$ and $p \in \mathrm{Conv}(A)$, there exists $m \leq d + 1$ points $p_1, \ldots, p_m \in A$ (denoted by a Carathéodory set of $p$) such that $p \in \mathrm{Conv}(\{p_1, \ldots, p_m\})$.*

Finally, it is known that some functions, including the ReLU function, do not admit an $\varepsilon$-coreset of size $o(n)$ [82, 81]. Thus, we use a generalized form of what is known as the complexity measure of a set of points, which was first introduced in [81] and later leveraged in [76]. This measure is used to determine the complexity of a given set $P$ with respect to ReLU, and the coreset size theoretically.

**Definition 2.3** (Regression Complexity Measure). Let $P \subseteq \mathbb{R}^d \times \{1\}$, the regression complexity measure of $P$ is defined as $\mu(P) = \sup_{x \in \mathbb{R}^{d+1}} \frac{\sum_{q \in \{p \in P | p^T x \leq 0\}} |q^T x|}{\sum_{q \in \{p \in P | p^T x > 0\}} |q^T x|}$, where the denominator is $\geq 0$, and the last entry of every $p \in P$ is 1, reserved for the bias/intercept term.

### 2.3 Our Pruning Scheme

In what follows, we present our data summarization technique for ReLU on the dot product function. Then in Section 2.5, we discuss that our results can be easily extended to a wide family of activation functions including the Sigmoid function, as recently shown in [76]. First we present Algorithm 1, which serves as a stepping stone towards bounding the sensitivities.

**Overview of Algorithm 1.** The algorithm receives as input a set $P \subset \mathbb{R}^d$ whose rank is $r \in [d]$ and deterministically finds a subset $S \subseteq P$ which satisfies that for every $j \in [d]$, $X \in \mathbb{R}^{d \times j}$ and $v \in \mathbb{R}^d$, $\frac{\max_{p \in P} \|(p-v)X\|_1}{\max_{p \in S} \|(p-v)X\|_1} \leq r^{1.5}$. To do so, first, we find the affine hyperplane that $P$ lies on, followed by computing the low dimensional representation of $P$, denoted by $P'$; see Lines 1–2. Note that if $\mathrm{rank}(P) = d$, then we can either keep $P$ as it is (i.e., $P' := P$), or use dimensionality reduction tricks as detailed in Section 2.5. To compute the output $S$, we first bound the convex hull of $P'$ by its Löwner ellipsoid $E(G, c)$ in Line 5, followed by computing the dilated ellipsoid of $E(G, c)$, namely, $\frac{1}{r}(E - c) + c$. Let $V$ be the set of vertices of such ellipsoid; see Line 6. Now, for each point $v \in V$, we represent it as a convex combination of $r + 1$ points from $P'$ via Theorem 2.2, and store the union of such sets (each of size at most $r + 1$) of points into $K$ as done in Lines 7–9. For each point in $K$, we map it back to $\mathbb{R}^d$ to satisfy Lemma 2.5. To sum up Algorithm 1, we observe that the vertices can be used via canonical combinations with their dilated form to describe every point on the convex hull of the input data (in our end, it would the network's weights). Hence the Carathéodory set of these vertices from the input points lying on the convex hull can be further used to also represent points lying on the convex hull. This is the core idea which enable us in forming our $\ell_\infty$ coreset for any $\ell_\rho$ regression problem where $\rho \in (0, \infty)$.

We now discuss Algorithm 2 which is responsible for constructing an $\varepsilon$-coreset with respect to activation functions. Its input is a set $P \subset \mathbb{R}^d$ and a sample size $m \geq 1$.

**Overview of Algorithm 2.** First set $Q := P$. At each iteration $i$, the algorithm obtains a subset $S_i \subseteq Q$ as an output to a call to $\ell_\infty$-CORESET($Q$) as stated in Line 3 of Algorithm 1 such that for every $x \in \mathbb{R}^d$, it holds that $\frac{\max_{p \in P} |p^T x|}{\max_{p \in S_i} |p^T x|} \leq r^{1.5}$ with $r$ being the rank of $Q$. The sensitivity of each point in $S_i$ is bounded from above by $\frac{2d^{1.5}}{i}$ as stated in Lines 4–6. The idea behind these bounds lies in our proof of Theorem 2.6. The set $S_i$ is removed from $Q$, and this procedure is repeated with respect to $Q$ until the size of $Q$ is small enough. The obtained sensitivities are the ones needed for computing the pruning coresets. Finally, we utilize the sensitivity sampling framework of [9] to obtain the desired coreset; see Lines 13–14.

### 2.4 Analysis

In this section, we prove the correctness of our algorithms. The following lemma shows that for each point $p$ that is inside some convex hull $S$, its $\ell_1$ distance to any affine subspace is always bounded from above by $\ell_1$ distance from the same affine subspace, of some other point $q \in S$.

**Lemma 2.4.** *Let $d, \ell, m \geq 1$ be integers. Let $p \in \mathbb{R}^d$ and $A \subseteq \mathbb{R}^d$ be a set of $m$ points with $p \in \mathrm{Conv}(A)$ so that there exists $\alpha : A \to [0, 1]$ such that $\sum_{q \in A} \alpha(q) = 1$ and $\sum_{q \in A} \alpha(q) \cdot q = p$. Then for every $Y \in \mathbb{R}^{d \times \ell}$ and $v \in \mathbb{R}^\ell$, $\|(p - v)Y\|_1 \leq \max_{q \in A} \|(q - v)Y\|_1$.*

The following states the provable guarantees of Algorithm 1.

**Lemma 2.5** ($\ell_\infty$-coreset for $\ell_1$-regression). *Let $P \subseteq \mathbb{R}^d$ be a set of points, and $r$ be the rank of $P$. Let $j \in [d-1]$ and let $S$ be the output of a call to $\ell_\infty$-CORESET$(P)$. Then (i) $|S| \in O\left(r^2\right)$, and (ii) for every $X \in \mathbb{R}^{d \times j}$ and $v \in \mathbb{R}^d$, $\frac{\max_{q \in P} \|(q-v)X\|_1}{\max_{q \in S} \|(q-v)X\|_1} \in \left[1, 2r^{1.5}\right]$.*

The following theorem states our main result.

**Theorem 2.6** (ReLU $\varepsilon$-coreset). *Let $P \subseteq \mathbb{R}^d$, $\varepsilon, \delta \in (0,1)$, $r = \mathrm{rank}\,(P)$, $\mu\,(P)$ be as in Definition 2.3, and let $m \in O\left(\frac{\mu(P)r^{3.5}\log n}{\varepsilon^2}\left(d\left(\log\left(\mu\left(P\right)r\log n\right)\right) + \log\left(\frac{1}{\delta}\right)\right)\right)$. Let $(C, w)$ be the output of a call to CORESET$(P, m)$; see Algorithm 2. Then, with probability at least $1 - \delta$, $(C, w)$ is an $\varepsilon$-coreset for $\left(P, \mathbb{R}^d, \mathrm{ReLU}\,(\cdot)\right)$; see Definition 1.1.*

**Time analysis.** Letting $r$ be the rank of $P$, the time complexity of Algorithm 1 can be dissected to two main parts: (i) Computing the Löwner ellipsoid in $O(nr^2 \log n)$ time using the method proposed in [98] and (ii) computing the Carathédory set in $O(nr + r^4 \log n)$ time via [70]. Since $V$ can contain up to $O(r^2)$ points, the overall time for Algorithm 1 is $O(nr^2 \log n + nr^3 + r^6 \log n)$. As for Algorithm 2, it takes $O(n^2 r^2 \log n + nr^4 \log n)$. Indeed, as explained in Section 2.5, a dimensionality reduction algorithm may be applied to improve the run time (reducing the $r^6$ factor). Furthermore, the run time of our algorithm can be improved, using the merge-and-reduce tree from the literature of coresets to reduce the $n^2$ terms to $n \log n$, i.e., the running time can be reduced to $O(n \log^2(n)r^2 + nr^4)$. For a data-independent provable method, this running time is reasonable.

**Our advantages over previous results.** Our coreset supports different activation functions without the need to change the sensitivity that much. Specifically, it will only be multiplied by some scalar, unlike previous coresets where different losses impose drastically different sensitivities/leverage scores and algorithms. This is since our coreset unlike other coresets is in its essence a framework of coresets for different $\ell_\rho$ losses, as it can be used as is for different $\ell_\rho$ losses and yet still attain $\epsilon$-approximation. In addition, when the rank of the input points is small, then our method outperforms previous methods. If the input data is of full rank, previous methods obtain faster coresets construction.

**On the boundness of the regression complexity measure.** First of all, there exists an example where the complexity measure is unbounded, e.g., consider a set of points distributed evenly on a unit ball. In this case, you can always find a point where a hyperplane separating it from the rest of the points can be found such that the one half-space of this hyperplane contains only this point while the other half-space contains the rest of the points. This leads to an infinite complexity measure. Such an example is also mentioned in [82], when assessing the hardness of generating multiplicative-approximation coresets for ReLU functions.

Theoretically, the complexity measure is influenced by how free can the bias term be (the last entry of x); see Definition 2.3. This term is the only thing that can ensure that one point can be separated from the rest in the sense of finding a separating hyperplane, leading to an infinite complexity measure. Bounding on this term, leads to bounded complexity measure from a theoretical point of view.

In the context of model pruning, from the perspective of the complexity measure, the model's weights are the input denoted by a matrix $P \in \mathbb{R}^{n \times (d+1)}$, while the query is now $\mathbb{R}^d \times \{1\}$. Thus the complexity measure is now $\mu\,(P) := \sup_{x \in \mathbb{R}^d \times \{1\}} \frac{\sum_{q \in \{p \in P | p^T x \leq 0\}} |q^T x|}{\sum_{q \in \{p \in P | p^T x > 0\}} |q^T x|}$. With this in mind, we observe that the complexity measure is now an instance of the complexity measure used in [76]. The complexity measure now relies entirely on the structure of the model's weights, where the goal is to find the largest ratio between the sum of the absolute of the values inside the rectified neurons prior to applying the rectification, and the sum values of non-rectified neurons. To bound this measure, we can use a variant of the algorithm described in the proof of Theorem 3 in [81].

## 2.5 Extensions

Our suggested scheme can be extended to support many other variants of the pruning problem.

**Various activation functions.** Our result can be extended to a family of activation functions called "Nice hinge functions"; see Definition D.1. Let $(P, X, \phi)$ be a query space, where $\phi$ is a "Nice hinge functions". To bound the sensitivity of a point $p$, we first bound the nominator of $s(p)$ by proving that $\forall x \in X : \phi(p^T x) < |p^T x|$. For bounding the denominator from below, recently [76] proved that $\forall x \in X : \sum_{p \in P} \mathrm{ReLU}\,(p^T x) \lesssim \sum_{p \in P} \phi(p^T x)$; see full detail in Section D.3.

**Weighted Input.** In the context of deep learning, the output of each neuron is multiplied with a scalar which brings the necessity of having the ability to deal with weighted set of points. Algorithm 2 can be extended easily to the case as generously detailed in Section D.1 in the Appendix.

**Dimensionality reduction.** All coreset-based pruning methods rely heavily on the dimensionality of the model's layers, as well as our method. To ensure sufficient pruning ratio, we apply either PCA, TSNE, MDS, or the JL transform on the weights of each layer prior to generating its coreset.

**From weight to neuron pruning.** Most coreset-based pruning methods, e.g., [5, 82], first provide a scheme for (provable) weight pruning, which is then used as a stepping stone towards pruning neurons as follows. [5] first suggested coreset-based neuron pruning via the use of a generated controled set of queries to evaluate the importance of weights. Any neuron that has a maximal activation value lower or equal to zero, will be pruned from the network as its impact on the rest of the neurons is minimal. On the other hand [82] altered the definition of sensitivity such that it takes into account the sensitivity of a neuron in a layer $\ell$ with respect to all the neurons in the layer $\ell + 1$, which basically means that the sensitivity of each neuron is taken be the maximal sensitivity over every weight function (neuron in the next layer) defined by the layer. Hence, we follow the same logic for such method; see Section D.2 in the supplementary material.

**From neuron to filter pruning.** Convolutions can be expressed as matrix multiplications, which enables our method to prune filters from any model as done by [60, 82, 61].

## 3 Experimental Results

In this section, we study various widely used network architectures and benchmark data-sets. Following [82], to test the robustness of our methods on each of the neuron and filter pruning tasks independently, two sets of experiments are conducted. The first focuses on pruning neurons (Section 3.1) whereas the second focuses on pruning filters (Section 3.2), both via our coreset method.

**The setting.** In all experiments we report the *Pruning ratio* – the percentage of the parameters that were removed from the original mode. Here, *PR* stands for pruning ratio, *FR* stands for floating-point reduction ratio and *Err* – the percentage of misclassified test instances of our method compared to coreset-based pruning methods and more. *Baseline Err* is the error of the original uncompressed network, while *Pruned Err* is the classification error of the compressed model. In our experiment we compress and fine-tune the network once, no iterative pruning was applied, thus, the compared methods also satisfy this setting. Each experiment was conducted 5 times, in the tables, we report for our method the best error achieved and we highlight in parentheses next to it the average error and standard deviation across the 5 trails. In all of our experiments, the models are fine-tuned till convergence (after pruning). Implementation details are given in Section E in the Appendix.

**Software/Hardware.** Our algorithms were implemented in Python 3.6 [108] using Numpy [83], and Pytorch [84]. Tests were performed on NVIDIA DGX A100 servers with 8 NVIDIA A100 GPUs each, fast InfiniBand interconnect and supporting infrastructure.

**Baselines.** Our results are compared to (i) PFP [60], (ii) FT [58], (iii) SoftNet [34], (iv) ThiNet [68], and (v) PvC [82], (vi) Soft Pruning [34], (vii) CCP [85], (viii) FPGM [36], (ix) ThiNet-70, (x) ThiNet-50 [68], (xi) Pruning via Coresets (PvC) [82], (xii) Pruning from Scratch (PfS) [111], and (xiii) Rethinking the value of network pruning (Rethink) [65].

### 3.1 Neuron Pruning

We test our method on VGG16 [92] using CIFAR10 [49], and LeNet300-100 using MNIST [55].

**Discussion.** Table 1 present the results of LeNet300-100 and VGG16. Observe that in both architectures, our method outperformed the competing methods under the same compression scenarios. For example, we pruned roughly $90\%$ of the parameters of the LeNet-300-100 model while improving the accuracy of the original model. We witness a similar phenomena on the VGG16 model, where we pruned roughly $90\%$ of the parameters of the dense layers resulting in accuracy improvement. This confirms the insights in [5] that coresets help in improving the generalization properties of DNNs.

Table 1: Pruning of LeNet-300-100 architecture on the MNIST dataset and of VGG-16 on the CIFAR-10 dataset. Here, we report the compression with respect to the fully connected layers only.

| Model | Method | Baseline Err. (%) | Pruned Err. (%) | PR (%) |
|---|---|---|---|---|
| LeNet-300-100 | PFP | 1.59 | 2.00 | 84.32 |
| | FT | 1.59 | 1.94 | 81.68 |
| | SoftNet | 1.59 | 2.00 | 81.69 |
| | ThiNet | 1.59 | 12.17 | 75.01 |
| | PvC | 2.16 | 2.03 | 90 |
| | **Our method** (90) | **2.07** | **1.98 (2.02 ± 0.04)** | **90** |
| | **Our method** (92.6) | **2.07** | **2.64 (2.74 ± 0.1)** | **92.6** |
| | **Our method** (94.6) | **2.07** | **3.51 (3.58 ± 0.07)** | **94.6** |
| VGG-16 | PvC | 8.95 | 8.16 | 75 |
| | **Our method** | **5.9** | **5.9 (6.2 ± 0.3)** | **90** |

## 3.2 Filter pruning

We compressed the convolutional layers of (i) ResNet50 [33] on ILSVRC-2012 [18], (ii) ResNet56 [33], (iii) VGG19 [92] on CIFAR10 and (iv) VGG16 [92] on CIFAR10.

Table 2: Filter pruning results on different neural networks with respect to the CIFAR10 dataset.

| Model | Method | Baseline Err. (%) | Pruned Err. (%) | PR (%) | FR (%) |
|---|---|---|---|---|---|
| VGG-19 | PfS | 6.4 | 6.29 | 52 | NA |
| | Rethink | 6.5 | 6.22 | 80 | NA |
| | Structured Pruning | 6.33 | 6.20 | 88 | NA |
| | PvC | 6.33 | 6.02 | 88 | NA |
| | **Our method** (88) | **6.33** | **5.85 (6.03 ± 0.18)** | **88** | NA |
| | **Our method** (91.28) | **6.33** | **6.23 (6.35 ± 0.12)** | **91.28** | NA |
| VGG-16 | ThiNet | 7.11 | 9.24 | 63.95 | 64.02 |
| | FT | 7.11 | 8.22 | 80.09 | 80.14 |
| | SoftNet | 7.11 | 7.92 | 63.95 | 63.91 |
| | PFP (94) | 7.11 | 7.61 | 94.32 | 85.03 |
| | PFP (87) | 7.11 | 7.17 | 87.06 | 70.32 |
| | PFP (80) | 7.11 | 7.06 | 80.02 | 59.21 |
| | **Our method** (95.32) | **7.11** | **7.31 (7.55 ± 0.24)** | **95.32** | **85.09** |
| | **Our method** (87) | **7.11** | **6.63 (6.76 ± 0.13)** | **87** | **68.2** |
| | **Our method** (79.53) | **7.11** | **6.3 (6.38 ± 0.08)** | **79.53** | **59.14** |
| ResNet56 | ThiNet | 7.05 | 8.433 | 49.23 | 49.74 |
| | Channel Pruning | 7.2 | 8.2 | N/A | 50 |
| | AMC | 7.2 | 8.1 | 64.78 | 50 |
| | CCP | 6.5 | 6.42 | 57 | 52.6 |
| | PvC | 6.21 | 7.0 | 55 | N/A |
| | **Our method** | **6.61** | **7.26 (7.56 ± 0.3)** | **63.95** | **50** |

Table 3: Filter pruning of ResNet50 on ImageNet (ILSVRC-2012).

| Method | Baseline Err. (%) | Pruned Err. (%) | PR (%) | FR (%) |
|---|---|---|---|---|
| PFP | 23.87 | 24.79 | 44.04 | 30.05 |
| Soft Pruning | 23.85 | 25.39 | 49.35 | 41.80 |
| CCP | 23.85 | 24.5 | 56 | 48.8 |
| FPGM | 23.85 | 25.17 | 62 | 53.5 |
| ThiNet-70 | 27.72 | 26.97 | 33.72 | 36.78 |
| ThiNet-50 | 27.72 | 28.0 | 51.5 | 55.82 |
| PvC | 23.78 | 25.11 | 62 | N/A |
| **Our method** | **23.78** | **24.87 (25.07 ± 0.2)** | **62** | **61.5** |
| **Our method** (18.01) | **23.78** | **24.1 (24.2 ± 0.1)** | **18.01** | **10.82** |

**Filter pruning of DNNs on CIFAR10 and ImageNet (ILSVRC-**2012**).** For Cifar10, we used PyTorch implementations of VGG19 and VGG16. We compressed both models using our approach by different compression rates as shown in Table 2 with a comparison to other methods. For ImageNet, we compressed the baseline model of ResNet50 [33] roughly by $62\%$ in terms of number of parameters. Table 3 provides comparison between our method and other baselines.

**Discussion.** As can be seen in Tables 1, 2, and 3, our method either outperforms the competing methods or achieves comparable results. As for the coreset methods, our algorithm achieves better result than all of them in this setting, e.g., we compressed $62\%$ of ResNet50 trained on ImageNet while incurring $1.09\%$ drop in accuracy, improving the recent coreset result of PvC [82] for the same compression ratio, while PFP [60] compressed $44.04\%$ to achieve comparable results.

## 4   Related work

DNNs can be compressed before training [97, 110, 57], during training [118, 115, 51], or after training [93]. Furthermore, such procedures may also be repeated iteratively [88]. As previously noted pruning can be categorized into structured and unstructured pruning.

**Unstructured pruning.** Weight pruning [56] techniques aim to reduce the number of weights in a layer while approximately preserving its output. Approaches of this type include the works of [54, 21, 39, 1, 62], where the desired sparsity is embedded as a constraint or via a regularizer into the training pipeline, and those of [31, 88, 30], where weights with absolute values below a threshold are removed. The approaches of [5, 6] use a mini-batch of data points to approximate the influence of each parameter on the loss function. Other data-informed techniques include [28, 63, 80, 79, 116]. A thorough overview of recent pruning approaches is given by [27, 8]. However, unlike our approach, weight-based pruning approaches generate sparse models instead of smaller ones thus requiring specialized hardware and sparse linear algebra libraries in order to speed up inference.

**Structured pruning.** Pruning entire neurons and filters directly shrinks the network leading to smaller storage requirements and improved inference-time performance on any hardware [59, 67]. Lately, these approaches were investigated in many papers [64, 59, 11, 36, 22, 46, 114, 113]. Usually, filters are pruned by assigning an importance score to each neuron/filter, either solely weight-based [37, 35] or data-informed [116, 60], and removing those with a score below a threshold. The procedure can be embedded into an iterative pruning scheme [88] that requires potentially expensive retrain cycles.

**Tensor decomposition.** Some of the work in DNN compression entails decomposing the layer into multiple smaller ones, e.g., via low-rank tensor decomposition [19, 41, 74, 48, 96, 40, 2, 105, 117, 53, 61]. Other approaches to tensor decomposition include weight sharing, random projections, and feature hashing [112, 3, 91, 12, 13, 107]. However, such techniques usually require expensive approximation algorithms or use heuristics since tensor decomposition is generally NP-hard.

**Coresets.** In the recent years, coresets got increasing attention, and where leveraged to compress the input datasets of many machine learning algorithms, improving there performance, e.g., regression [72, 38, 81, 47, 103], decision trees [44], matrix approximation [26, 70, 25, 89, 73], data discretization [75], clustering [24, 29, 66, 4, 45, 90, 106], $\ell_z$-regression [16, 17, 94], *SVM* [32, 101, 99, 100, 102], deep learning models [69, 5, 60] and even for path planning in the field of robotics [104]. For extensive surveys on coresets, we refer the reader to [23, 86, 43, 71].

## 5   Conclusions and Future Work

In this paper, we provided a coreset-based pruning technique that hinges upon a combination of tools from convex geometry, while achieving SOTA results with respect to coreset-based structured pruning approaches on a variety of networks. Our main improvement is that our coreset is (training) data-independent and assumes a single assumption on the models weights.

**Future work includes** (i) suggesting a coreset based budget allocation framework, to determine the (optimal) per layer prune ratio while achieving an overall desired compression rate, (ii) extending our coreset technique to other layers such as attention layers, and (iii) bridging the gap between coreset based pruning approaches and tensor-decomposition methods, as both techniques are theoretically supported by bounding the approximation error given specific compression rate, we can leverage these

bounds to formulate the compression problem as an optimization problem which iterates between the two approaches to search for the local minimum.

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
