## A Computing the Carathéodory set

**Overview of Algorithm 4.** First, a convex combination of $v$ with respect to $P$ is formulated as a linear programming problem. This is done by reformulating the input set of points $P$ as a matrix denoted as $A \in \mathbb{R}^{(d+1) \times n}$ (see Line 1). We then formulate the goal vector $b \in \mathbb{R}^{d+1}$ to be $v$ concatenated with an entry of 1 which serves to make sure that the solution to

$$\begin{aligned}\underset{x \in \mathbb{R}^{n+1}}{\text{minimize}} \quad & \mathbf{1}_{n+1}^T x \\ \text{subject to} \quad & Ax = b, \\ & x_i \in [0,1] \quad \forall i \in [d]\end{aligned}$$

satisfies that $\sum_{i \in [n]} x_i P_i = v$ and $\sum_{i \in [n]} x_i = 1$. Solving this problem takes roughly $O^*\left(n^{\omega + o(1)}\right)$ where $\omega$ is the matrix multiplication exponent as elaborated in [15]; see Lines 2–3. We observe that $x$ from Line 3 might be dense, i.e., the number of non-zero entries exceeds $d + 1$. To ensure that we have at max $d + 1$ non-zero entries, we use Algorithm 1 of [70] which aims to find a set of $d + 1$ points, where their weighted average is the desired $v$ given the initial weight vector $x$; see Line 4.

---

**Algorithm 3:** CARATHEODORY-SET $(v, P)$

**Input** : A point $v \in \mathbb{R}^d$ and a set $P \subseteq \mathbb{R}^d$ of $n$ points
**Output** : A subset $C \subseteq P$ of at max $d + 1$ points such that $p \in \text{Conv}(C)$

1 $A := \left[ \begin{bmatrix} P_1 \\ 1 \end{bmatrix}, \begin{bmatrix} P_2 \\ 1 \end{bmatrix}, \ldots, \begin{bmatrix} P_n \\ 1 \end{bmatrix} \right]$ /* $P_i$ here denotes the $i$th point in $P$ */

2 $b := \begin{bmatrix} v \\ 1 \end{bmatrix}$

3 $x := \underset{\substack{x \in [0,\infty)^d \\ Ax=b}}{\arg \min}\ \mathbf{1}_d^T x$ // $\mathbf{1}_d$ denotes a $d$ dimensional vector of 1s

4 $C := \text{FAST-CARATHEODORY-SET}\left(P, x, d^2 + 2\right)$ /* See Algorithm 1 of [70] */

5 **return** $C$

---

## B Coreset-Related Technical Details

**Definition B.1** (VC-dimension [9]). For a query space $(P, w, \mathbb{R}^d, f)$ and $r \in [0, \infty)$, we define

$$\text{ranges}(x, r) = \{p \in P \mid w(p) f(p, x) \leq r\},$$

for every $x \in \mathbb{R}^d$ and $r \geq 0$. The dimension of $(P, w, \mathbb{R}^d, f)$ is the size $|S|$ of the largest subset $S \subset P$ such that

$$\left|\left\{S \cap \text{ranges}(x, r) \mid x \in \mathbb{R}^d, r \geq 0\right\}\right| = 2^{|S|},$$

where $|A|$ denotes the number of points in $A$ for every $A \subseteq \mathbb{R}^d$.

### B.1 Sensitivity Sampling Missing Details

We want to use the sensitivity sampling framework to compute a coreset for a set of points $P$ in $\mathbb{R}^d$.

First, we need to bound the sensitivity of each point $p \in P$. The sensitivity pf a point $p \in P$ is defined as $s(p) = \sup_{x \in X} \frac{\phi(p,x)}{\sum_{q \in P} \phi(q,x)}$ where the denominator is not zero.

Hence, for every $p \in P$, we wish to compute a number $s'(p)$, such that $s'(p) \geq s(p)$. Once the bound $s'(p)$ on the sensitivity $s(p)$ of each point $p$ is computed, we define $T = \sum_{p \in P} s'(p)$ as the total sensitivity. Now, to obtain a coreset, we can sample points according to the distribution $s'(p)/T$, i.e., we sample $m > 0$ points from $P$, where at each sample, the point $p \in P$ is sampled i.i.d with probability $s'(p)/T$. We also re-weight the sampled points to obtain a coreset.

As the bound $s'(p)$ (on $s(p)$) is tighter, the total sensitivity $T$ gets smaller, and then the coreset size (required number of sampled points) gets smaller, and vice versa.

**Theorem B.2** (Restatement of Theorem 5.5 in [9]). *Let $P \subseteq \mathbb{R}^d$ be a set of $n$ points, $w : P \to [0, \infty)$ be a weight function , and let $f : P \times \mathbb{R}^d \to [0, \infty)$ be a loss function. For every $p \in P$ define the sensitivity of $p$ as*

$$\sup_{x \in \mathbb{R}^d} \frac{w(p)f(p, x)}{\sum_{q \in P} w(q)f(q, x)},$$

*where the sup is over every $x \in \mathbb{R}^d$ such that the denominator is non-zero. Let $s : P \to [0, 1]$ be a function such that $s(p)$ is an upper bound on the sensitivity of $p$. Let $t = \sum_{p \in P} s(p)$ and $d'$ be the VC dimension of the quadruple $(P, w, \mathbb{R}^d, f)$; see Definition B.1. Let $c \geq 1$ be a sufficiently large constant, $\varepsilon, \delta \in (0, 1)$, and let $S$ be a random sample of $|S| \geq \frac{ct}{\varepsilon^2} \left(d' \log t + \log \frac{1}{\delta}\right)$ i.i.d points from $P$, such that every $p \in P$ is sampled with probability $\frac{s(p)}{t}$. Let $v(p) = \frac{tw(p)}{s(p)|S|}$ for every $p \in S$. Then, with probability at least $1 - \delta$, $(S, v)$ is an $\varepsilon$-coreset for $(P, w, \mathbb{R}^d, f)$.*

## B.2 From Coresets to Approximating the Optimal Solution

In optimization problems (or machine learning in general), the goal is usually to find a query that minimizes (or maximizes) some cost function. In the context of coresets, the goal is to find a small weighted subset such that for a given cost function, the cost of applying any solution (hypotheses/query) on the coreset approximates the cost of applying the same solution on the whole data. Since a coreset approximates the cost of every query, we do note that in many cases, coresets are applied for approximating the optimal solution. Specifically, solving the desired optimization problem on the whole data can be a hard problem when the time needed for such a solution is either polynomial or exponential in the size of the whole data, or when the required memory is too high. In this case, coresets can be leveraged, by computing the the optimal solution of fitting an $\varepsilon$-coreset and applying it on the original data. If the computed coresets gives worst-case $(1 + \varepsilon)$-approximation error, then we provably $(1 + 4\varepsilon)$-approximation towards the optimal cost of solving the optimization on the whole data (the proof is very easy, it is done by applying the triangle inequality few times). In other words, we can solve the problem on the coreset to obtain a solution $x^*$, and then apply $x^*$ to the whole data giving a good approximation for solving the problem from the beginning on the whole data.

# C Proofs of Technical Results

## C.1 Proof of Lemma 2.4

**Lemma C.1.** *Let $d, \ell, m \geq 1$ be integers. Let $p \in \mathbb{R}^d$ and $A \subseteq \mathbb{R}^d$ be a set of $m$ points with $p \in \text{Conv}(A)$ so that there exists $\alpha : A \to [0, 1]$ such that $\sum_{q \in A} \alpha(q) = 1$ and $\sum_{q \in A} \alpha(q) \cdot q = p$. Then for every $Y \in \mathbb{R}^{d \times \ell}$ and $v \in \mathbb{R}^\ell$, $\|(p - v)Y\|_1 \leq \max_{q \in A} \|(q - v)Y\|_1$.*

*Proof.* Since we can write $p$ as the convex combination of points $q \in A$ with weight $\alpha(q)$, we have

$$\|p^T Y - v\|_1 = \left\| \left(\sum_{q \in A} \alpha(q)q^T\right) Y - v \right\|_1.$$

Moreover, we have $\sum_{q \in A} \alpha(q) = 1$, so we can decompose $v$ into

$$\|p^T Y - v\|_1 = \left\| \sum_{q \in A} \alpha(q) \left(q^T Y - v\right) \right\|_1.$$

By triangle inequality (or Jensen's inequality),

$$\|p^T Y - v\|_1 \leq \sum_{q \in A} \alpha(q) \|q^T Y - v\|_1 \leq \max_{q \in A} \|q^T Y - v\|_1.$$

$\square$

## C.2 Proof of Lemma 2.5

**Lemma C.2** ($\ell_\infty$-coreset for $\ell_1$-regression). *Let $P \subseteq \mathbb{R}^d$ be a set of points, and $r$ be the rank of $P$. Let $j \in [d-1]$ and let $S$ be the output of a call to $\ell_\infty$-CORESET$(P)$. Then (i) $|S| \in O(r^2)$, and (ii) for every $X \in \mathbb{R}^{d \times j}$ and $v \in \mathbb{R}^d$, $\frac{\max_{q \in P} \|(q-v)X\|_1}{\max_{q \in S} \|(q-v)X\|_1} \in [1, 2r^{1.5}]$.*

*Proof.* Let $Y, z, P', K, S$ and $\mathrm{map}$ be defined as in Algorithm 1. Since $P$ lies on a $r$-dimensional affine subspace, it holds that for every $p \in P$, $p = (p-z)YY^T + z$. Note that $Y \in \mathbb{R}^{d \times r}$ is an orthogonal matrix (i.e., $Y^TY$ is the identity matrix in $\mathbb{R}^{r \times r}$) and $z \in \mathbb{R}^d$ denotes the translation of the affine subspace that $P$ lies on.

**Claim (i).** $E(G, c)$ is the Löwner ellipsoid of $P'$, which has $2r$ vertices. Since $V$ is the set of vertices of the shrunk form of $E(G, c)$ that is contained in Conv $(P')$, each point from $V$ can be represented as convex combination of $r+1$ points from $P'$ by Carathéodory's Theorem. Then the number of points in $K$ is at most $2r(r+1)$, i.e., $|K| \in O(r^2)$. Thus, $|S| \in O(r^2)$ due to the fact that it can be constructed from $K$ through the use of $\mathrm{map}$.

**Claim (ii).** First put $X \in \mathbb{R}^{d \times j}$ and $v \in \mathbb{R}^d$, and let $p \in \arg\sup_{q \in P} \|(p-v)X\|_1$. Since $S \subseteq P$, it holds that $\frac{\max_{q \in P} \|(q-v)X\|_1}{\max_{q \in S} \|(q-v)X\|_1} \geq 1$. Let $a := zYY^T + z - v$, we have

$$\|(p-v)X\|_1 = \left\|\left((p-z)YY^T + z - v\right)X\right\|_1^T = \left\|\left(pYY^T + a\right)X\right\|_1 = \left\|\left(p'Y^T + a\right)X\right\|_1, \tag{1}$$

where the first equality holds since $\mathrm{rank}(P) = r$, the second holds by definition of $a$, and the last equality holds by the construction of $p' = pY$ at Line 2 of Algorithm 1.

Note that since $V$ is the set of vertices of $\frac{1}{r}(E(G, c) - c) + c$, by the definition of the Löwner ellipsoid,

$$V \subseteq \mathrm{Conv}\,(V) \subseteq \frac{1}{r}(E(G, c) - c) + c \subseteq \mathrm{Conv}\,(P') \subseteq E(G, c) \subseteq \mathrm{Conv}\left(r^{1.5}\,(V - c) + c\right).$$

Since Conv $(P')$ enclose Conv $(V)$, and is enclosed by Conv $\left(r^{1.5}\,(V - c) + c\right)$, then there exists a point $q \in \mathrm{Conv}\,(V)$ and $\gamma \in [0, 1]$ such that $p'Y^T = \gamma qY^T + (1 - \gamma)\left(r^{1.5}\,(q - c) + c\right)Y^T$, where by definition it holds that $r^{1.5}\,(q - c) + c \in \mathrm{Conv}\left(r^{1.5}\,(V - c) + c\right)$, and $p' = pY$.

By invoking Lemma 2.4, we obtain that

$$\left\|\left(p'Y^T + a\right)X\right\|_1 \leq \max\left\{\left\|\left(qY^T + a\right)X\right\|_1, \left\|\left(r^{1.5}\,(q - c) + c + a\right)Y^TX\right\|_1\right\}. \tag{2}$$

We note that

$$\left\|\left(qY^T + a\right)X\right\|_1 \leq \max_{\tilde{q} \in V}\left\|\left(\tilde{q}Y^T + a\right)X\right\|_1 \leq \max_{\tilde{q} \in K}\left\|\left(\tilde{q}Y^T + a\right)X\right\|_1, \tag{3}$$

where the first inequality follows from plugging $p := q$ and $A := V$ into Lemma 2.4, and the second inequality holds similarly since every point $V$ lies in Conv $(K)$. By invoking triangle inequality, we obtain that

$$\left\|\left(r^{1.5}\,((q - c) + c)Y^T + a\right)X\right\|_1 = \left\|\left(r^{1.5}qY^T + \left(1 - r^{1.5}\right)cY^T + a\right)X\right\|_1 \tag{4}$$
$$= \left\|\left(r^{1.5}qY^T + r^{1.5}a + \left(1 - r^{1.5}\right)cY^T + \left(1 - r^{1.5}\right)a\right)X\right\|_1$$
$$\leq r^{1.5}\left\|\left(qY^T + a\right)X\right\|_1 + \left(r^{1.5} - 1\right)\left\|\left(cY^T + a\right)X\right\|_1,$$

where the first equality follows by a simple rearrangement, and the second holds since $r^{1.5}a + \left(1 - r^{1.5}\right)a = a$. Observe that $c \in \mathrm{Conv}\,(V)$. Hence, by Lemma 2.4,

$$\left\|\left(cY^T + a\right)X\right\|_1 \leq \max_{\tilde{q} \in K}\left\|\left(\tilde{q}Y^T + a\right)X\right\|_1. \tag{5}$$

By the construction of $S$, it holds that for every $p \in S$, $pY \in K$. Thus, combining (2), (3), (4) and (5) yields $\frac{1}{2r^{1.5}}\left\|\left(p'Y^T + a\right)X\right\|_1 \leq \max_{\tilde{q} \in K}\left\|\left(\tilde{q}Y^T + a\right)X\right\|_1 = \max_{\tilde{q} \in S}\left\|\left(\tilde{q}YY^T + a\right)X\right\|_1 = \max_{\tilde{q} \in S}\left\|\left(\tilde{q} - v\right)X\right\|_1$ where the last equality holds by (1). This concludes Lemma 2.5. $\qquad\square$

## C.3 Proof of Theorem 2.6

*Proof.* For space constraints let $\phi$ denote the ReLU function. To obtain a coreset, we first need to bound the sensitivity of each $p \in P$. Put $p \in P$ and let $x \in \arg\sup\limits_{x' \in \mathbb{R}^d} \frac{\phi(p^T x')}{\sum_{q \in P} \phi(q^T x')}$ where the supremum is over every $x' \in \mathbb{R}^d$ such that the denominator is not zero. Observe that

$$\frac{\sum\limits_{q \in P} |q^T x|}{\sum\limits_{q \in P} \phi(q^T x)} = \frac{\sum\limits_{q \in P} \phi(q^T x) + \sum\limits_{q \in P} \phi(-q^T x)}{\sum\limits_{q \in P} \phi(q^T x)} = 1 + \frac{\sum\limits_{q \in P} \phi(-q^T x)}{\sum\limits_{q \in P} \phi(q^T x)} \leq 1 + \mu(P),$$

where the last inequality follows from Definition 2.3. Thus

$$\sum_{q \in P} \phi(q^T x) \geq \frac{1}{1 + \mu(P)} \sum_{q \in P} |q^T x|. \tag{6}$$

Let $\beta = (1 + \mu(P))$. We next observe that $\frac{\phi(p^T x)}{\sum_{q \in P} \phi(q^T x)} \leq \frac{|p^T x|}{\sum_{q \in P} \phi(q^T x)} \leq \beta \frac{|p^T x|}{\sum_{q \in P} |q^T x|}$, where the first inequality holds by properties of $\phi$, and the second is by (6). Hence the sensitivity of $p$ is bounded by

$$s(p) = \frac{\phi(p^T x)}{\sum_{q \in P} \phi(q^T x)} \leq \beta \frac{|p^T x|}{\sum_{q \in P} |q^T x|}. \tag{7}$$

Let $i$ be the iteration counter from Algorithm 2 as defined in Line 1, used in Line 5 and incremented in Line 7. The idea follows that of [109] where points being discarded from $P$ at lower levels (smaller $i$'s) have higher sensitivity. This notion also resembles that of the "Onion sampling" of [45]. Now, assume that $p \in S_i$ at iteration $i$ of the while loop (Line 3 of Algorithm 2). In this case, observe that by plugging $P := Q \setminus \bigcup_{\hat{i}=1}^{i-1} S_{\hat{i}}$, $j = 1$ and $v = -b\frac{x}{\|x\|_2}$ into Lemma 2.5, we obtain a subset $S_i \subseteq Q$ such that

$$\max_{q \in P} |q^T x| \leq \max_{q \in S} 2r^{1.5} |q^T x|. \tag{8}$$

Thus for every $q \in S_i$,

$$\frac{s(p)}{(1 + \mu(P))} \leq \frac{|p^T x|}{\sum_{q \in P} |q^T x|} \leq \frac{|p^T x|}{\sum_{\hat{i}=1}^{i} \max_{q \in S_{\hat{i}}} |q^T x|} \leq 2r^{1.5} \frac{|p^T x|}{\sum_{\hat{i}=1}^{i} |p^T x|} = \frac{2r^{1.5}}{i},$$

where the first inequality is by (7), the second inequality follows from the observation that $\left\{ \arg\max_{q \in S_{\hat{i}}} |q^T x| \right\}_{\hat{i}=1}^{i} \subseteq P$ and the last inequality holds by (8). Hence, we have obtained a bound on the sensitivity of each point $p \in P$. As for the total sensitivity, we observe that $t = \sum_{p \in P} s(p) \in O\left((1 + \mu(P)) r^{3.5} \log n\right)$. Theorem B.2 states that to obtain an $\varepsilon$-coreset with probability at least $1 - \delta$, the sample size $m$ must be $O\left(\frac{\mu(P) r^{3.5} \log n}{\varepsilon^2} \left(d\left(\log\left(\mu(P) r \log n\right)\right) + \log\left(\frac{1}{\delta}\right)\right)\right)$. $\square$

## D Extension

### D.1 Handling Weighted Sets of Points

Similarly to [5, 60], we split the input data $P$ into two sets $P_+, P_- \subseteq P$ such that $P_+ = \{p \in P | w(p) \geq 0\}$ while $P_- = \{p \in P | w(p) < 0\}$. Following this step we call Algorithm 4 for each of the two sets with corresponding weights and corresponding sample sizes. To account for proper sample sizes, we split our theoretical bound of the required sample size for generating $\varepsilon$-coreset into two terms for both $P_+$ and $P_-$ respectively, i.e., we formulate $m = m_+ + m_-$ where $m_+ = \frac{|P_-|}{|P|} m$ (similarly for $m_-$). Hence, we obtain an $\varepsilon$-coreset for each of the query spaces $(P_+, w, \mathbb{R}^d, \phi)$ and $(P_-, w, \mathbb{R}^d, \phi)$.

## D.2 From Weight to Neuron Pruning

Most coreset-based pruning methods, e.g., [5, 82], first provide a scheme for (provable) weight pruning, which is then used as a stepping stone towards pruning neurons as follows. To prune neurons from a layer, post to computing the coreset-based weight pruning for each neuron, ideally we would have that at certain layer, for all neurons, the generated coreset contains the same set of neurons from previous layers, which in this case we can remove the neurons which are not in the coreset. However, such scenario is almost implausible. To deal with such problem, we discuss two ways to do so. The first method to deal with such problem is inspired by the technique used in [82] which alters the definition of sensitivity such that it takes into account the sensitivity of a neuron in a layer $\ell$ with respect to all the neurons in the layer $\ell + 1$, basically the sensitivity of each neuron is taken be the maximal sensitivity over every weight function (neuron in the next layer) defined by the layer. Hence, we follow the same logic for such method, more details .

---

**Algorithm 4:** GENERALIZED-CORESET $(P, w, m)$

---

**input** : A set $P \subseteq \mathbb{R}^d$ of $n$ points, a weight function $w(p) : P \to [0, \infty)$ and a sample size $m$
**output :** A weighted set $(C, u)$

1   $Q := P$, $i := 1$, $C := \emptyset$
2 **while** $|Q| \geq 2\text{rank}(Q)^2$ **do**
3     $Q' := \{w(q)q | q \in Q\}$
4     $\text{map}_w : Q' \to Q$ a map that maps from $Q'$ to $Q$
5     $S_i := \ell_\infty\text{-CORESET}(Q')$
6     **for** *every* $p \in S_i$ **do**
7        $s\left(\text{map}_w(p)\right) := \frac{2r^{1.5}}{i}$
8     **end**
9     $Q := Q \setminus \{\text{map}_w(q) | q \in S_i\}$, $i := i + 1$
10 **end**
11 **for** *every* $p \in Q$ **do**
12     $s(p) := \frac{2r^{1.5}}{i}$
13 **end**
14 $t := \sum_{p \in P} s(p)$
15 $C := $ an i.i.d sample of $m$ points from $P$, where each $p \in P$ is sampled with probability $\frac{s(p)}{t}$.
16 $u(p) := \frac{tw(p)}{m \cdot s(p)}$ for every $p \in C$
17 **return** $(C, u)$

---

## D.3 Other Activation Functions

[76] recently showed that there exists a family of functions $\mathcal{F}$ called "Nice hinge functions" such that for any query $x \in \mathbb{R}^d$ and a set of points $P \subseteq \mathbb{R}^d$, for any $\phi \in \mathcal{F}$, it holds that $\frac{\sum_{p \in P} \phi(p^T x)}{\sum_{q \in P} \text{ReLU}(q^T x)}$ is bounded from below. Formally speaking, below is the definition of a "nice hinge function".

**Definition D.1** (Restatement of Definition 7 of [76]). We call $f : \mathbb{R} \to [0, \infty)$ an $(L, a_1, a_2)$-nice hinge function if for a fixed constant $L$, $a_1$ and $a_2$,

    (i) $f$ is $L$-Lipschitz,

    (ii) $|f(z) - \text{ReLU}(z)| \leq a_1$ for all $z$, and

    (iii) $f(z) \geq a_2$ for all $z \geq 0$.

As noted by [76], the hinge and log losses are $(1, 1, 1)$-nice and $(1, \ln 2, \ln 2)$-nice hinge functions respectively. Similarly, it is easy to show that the activation function $\phi(x) = \ln(1 + e^x)$ is

$(1, \ln 2, \ln 2)$-nice hinge functions. Following the same steps applied by [76], we obtain that

$$\sum_{p \in P} \phi\left(p^T x\right) \geq \min\left\{\frac{a_2}{2a_1}, \frac{1}{2}\right\} \frac{\sum_{q \in P} |q^T x|}{\mu(P) + 1},$$

where $\phi(\cdot)$ is a $(L, a_1, a_2)$ where $a_2$ is assumed to be positive.

Unlike the $\mathrm{ReLU}$ activation function, to support for other activation functions, we need to restrict our query space to contain queries such that $\forall p \in P : \phi\left(p^T x\right) \leq r |p^T x|$ where $r$ denotes the rank of $P$. Let $X'$ denote the set of all such queries.

Hence, under this additional assumption, we obtain that for every $p \in P$ and $x \in X'$

$$\frac{\phi\left(p^T x\right)}{\sum_{q \in P} \phi\left(q^T x\right)} \leq \frac{(1 + \mu(P)) \phi\left(p^T x\right)}{\min\left\{\frac{a_2}{2a_1}, \frac{1}{2}\right\} \sum_{q \in P} |q^T x|} \leq \frac{(1 + \mu(P)) r |p^T x|}{\min\left\{\frac{a_2}{2a_1}, \frac{1}{2}\right\} \sum_{q \in P} |q^T x|}.$$

Following the same steps done at the proof of Theorem 2.6, we can generate an $\varepsilon$-coreset with respect to $(P, \phi, X')$.

# E    Implementation Details

First observe that the *map* function in Line 3 of Algorithm 1 is hard to implement if all we have is $Y$ and $P'$ (see Lines 1–2) due to the fact that when the rank of $P$ is not $d$, then $Y$ becomes a singular matrix. A way around such problem (practically speaking yet also theoretically sound) is to reformulate $P$ and $P'$ as matrices, where our $\ell_\infty$-coreset will now be regarded as a set of indices of the rows selected as the desired coreset. **Note** that while we relied on an "accurate" measure of the rank of points in Algorithm 1, in our experiments, we used the rank function from *Numpy*, and still produced favorable results. Furthermore, our algorithms work also when the input has full rank. In addition, we can still obtain an $\varepsilon$-coreset when using approximated algorithms for the rank computation problem, where the error associated with our coreset may increase. In this case, we can increase our coreset size to reduce our approximation error to be the original desired error.

# F    Complexity Measure - Clarification

First, Note that while the complexity measure was first defined for construction of coresets with respect to the logistic regression problem, it also has been used for the ReLU regression problem (minimizing the sum of ReLU losses) [76].

The complexity is expected to be small other than in some cases [81, 76]. We also operate under the same assumption, i.e., the complexity measure is reasonably small.

Specifically speaking, when given a set $P$ (expressing our neurons) containing points in $\mathbb{R}^3$ (for example), such that point in $P$ lie on a 2-dimensional affine subspace parallel to the $xy$-plane, notice that in our setting, the complexity measure is defined as the maximal value of an optimization problem involving our vectors and a set of queries $X := \mathbb{R}^2 \times \{1\}$.

The existing of a hyperplane whose normal in $X$ such that one point from $P$ can be separated from the rest of the points in $P$, leads to large complexity measure.

In Figure 3, a clear separation can be made between one point and the rest leading to two sets of points: The first set contains one single point that has a positive dot product with the normal $x$ to the separating hyperplane, while the other set contains the remaining point each with negative dot product with $x$. This leads to a large complexity measure, and as the separating hyperplane gets closer and closer to the set containing the single point, the complexity measure increases, as it can tend to infinity.

On the other hand, when one can not separate a single point or minimal set of points from the rest of the data, we expect the complexity measure to be small; see Figure 4.

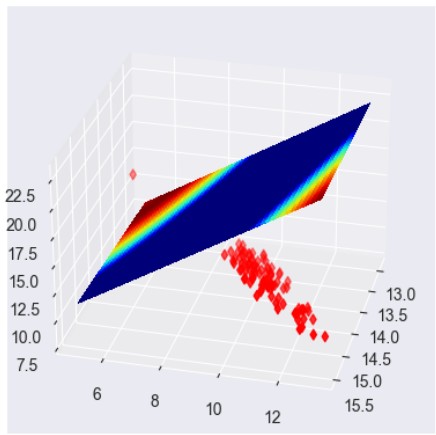

Figure 3: Linearly separable data leading to sufficiently large complexity measure.

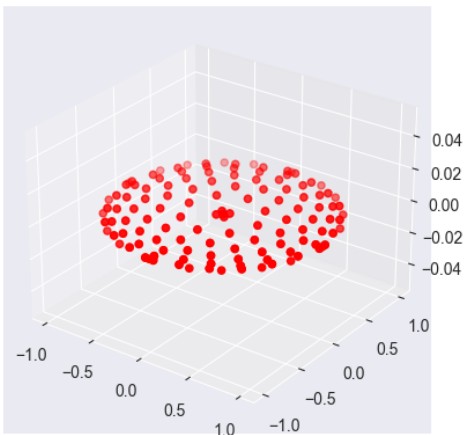

Figure 4: Non-linearly separable data leading to sufficiently small complexity measure.

Notice that in Figure 4, the input data is centered around the origin, which means that the data is not linearly separable. Thus leading to small complexity measure, since $x$ represents the normal to a hyperplane emerging from the origin.

In fact, during our experiments, the complexity measure in the case of LeNet300-100 was around 15 when the input data (matrix representing the neurons) had 300 rows (points).

It is common in coreset literature from a practical point of view, sample sizes that are smaller than the bound on the coreset size are being used. This is due to the fact that such bounds are pessimistic in nature.

This motivated the choice of not incorporating the complexity measure in our sample size nor the sensitivity sampling since such a term will be eliminated when computing the sampling probability; see Theorem B.2. Our experiments confirmed such an observation, i.e., our coreset lead to favorable results when the complexity measure was not incorporated in our computations, or when the sample size was much smaller than the bound on the coreset size.

The appendix in the supplementary material has been modified in light of this.

# G   Additional Experiments

In all of our experiments, our hyper-parameters were drawn from [60].

## G.1   The effect of fine-tuning

In this experiment, we aim to show the effect of fine-tuning on our compressed model. Specifically, Figure 5 shows VGG19's network accuracy over fine-tuning, where we start better than previous methods, followed by a slow incline in accuracy until we outperform previous models (around epoch 18).

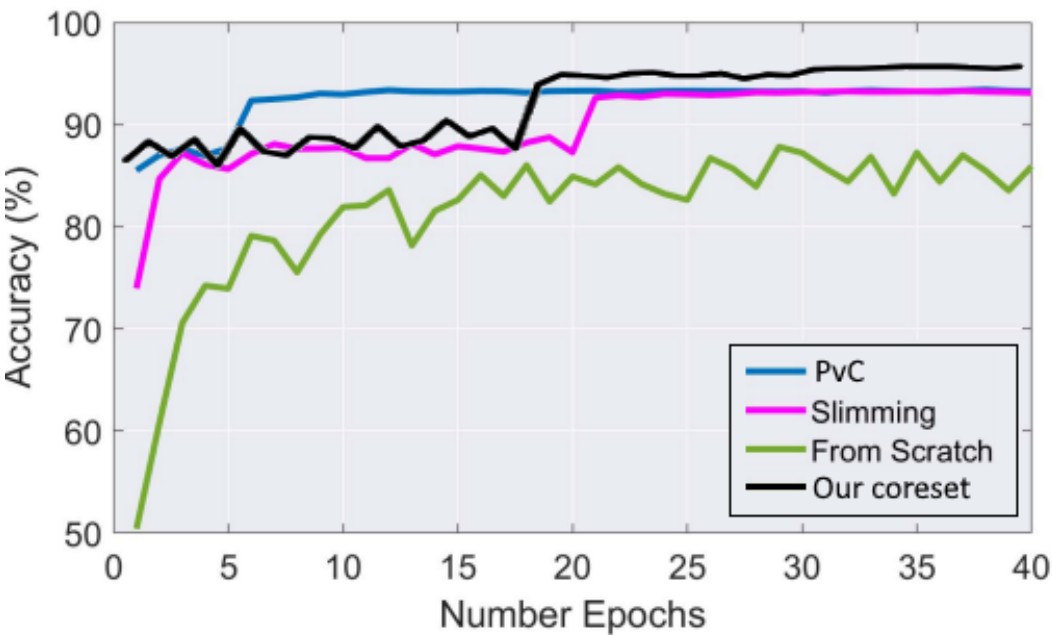

Figure 5: Accuracy of our proposed framework in comparison to previous methods and training the pruned network from scratch. The results above reflect the accuracy of VGG19 on CIFAR10.

## G.2   Sensitivity based distribution

At Figure 6, we plot the sensitivity distribution of our sampling method in comparison to the sampling probabilities achieved by [82]. Our advantage lies in the observation that our induced probability distribution entails longer tails, i.e., important point are scarce.

## G.3   Comparison with PvC

In this experiment, we aim to show the efficacy of our approach against that of [82]. We considered a single neuron in LeNet-300-100 where we computed the average additive error of the cost of the coreset from the cost of taking all the samples (neurons from previous layer), over set of 1000 queries. As shown in Figure 7, for very small coreset sizes, PvC [82] attains smaller error, however as we increase the sample size, our coreset outperforms that of [82].

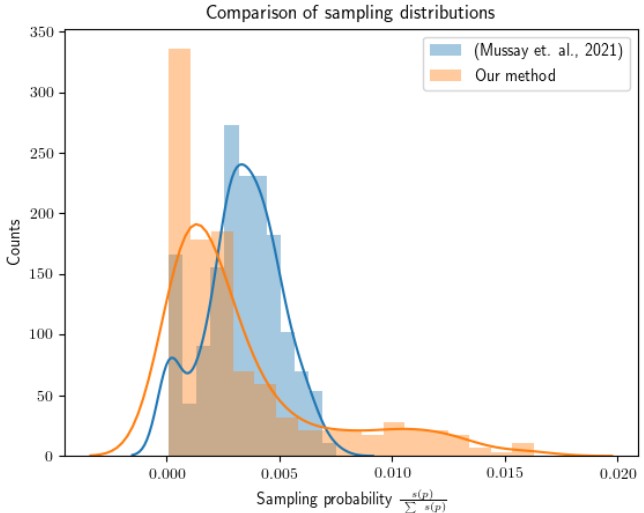

Figure 6: A comparison against [82] with respect to the distribution of the sampling probabilities of weights of a single neuron at some layer of LeNet-300-100. Here the $x$-axis denotes the sampling probability of points, while the $y$-axis presents the number of points with certain probability.

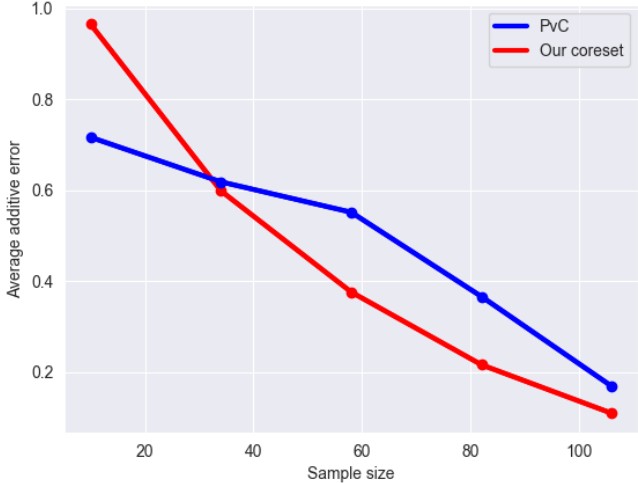

Figure 7: Average additive error of our coreset and that of [82] on LeNet-300-100.