# OpenReview forum: "Pruning Neural Networks via Coresets and Convex Geometry: Towards No Assumptions"
_NeurIPS.cc/2022/Conference — NeurIPS 2022 Accept_

### Official Review · Reviewer_4TWt · 2022-07-07

**Rating:** 6
**Confidence:** 4
**Soundness:** 3 good
**Presentation:** 3 good
**Contribution:** 3 good

**Summary:**

**Summary of results**
The main scope is compression of neural networks with coreset: How to remove neurons or filters after training such that we achieve almost the same performance. The paper proposes an $\ell_{\infty}$ coreset for $\ell_1$ regression (in Lemma 2.7). This coreset is exploited for pruning a single-layer neural network with ReLU activations with all-one output weights (details in Definition 2.2). Compared to existing results in the literature,  the proposed theoretical guarantee is data-independent without explicit assumptions on the norm of inputs. Experimentally comparisions have been conducted on standard neural networks to compare the proposed coreset-based pruning with baselines.

**Comments on relevance and importance of the scope**
Pruning neurons not only enhance the storage and computational complexity of neural networks but also may enhance the generalizability of the nets. Thus, it is a core topic in neural computing.





**Questions:**

1. There are coreset designed for l1 regression with similar techniques:(i) Lowner ellipsoid and (ii) Caratheodory set (see, for example, the survey on Sampling Algorithms and Coresets for $\ell_p$ Regression). I wondered what the novelty of the proposed algorithm compared to these exciting methods for l1 regression.
2. Can authors validate boundedness of regression complexity measure for simple cases?
3. Is it possible to characterize a worse case for the regression complexity measure?


**Limitations:**

This project does not have negative social impacts.

**Strengths And Weaknesses:**


**Advantages**
1. Compared to existing results, the coreset size is data-independent up to the regression complexity measure.
2. The proposed algorithm handles low-rank weight matrices that are essential in pruning neural networks.
3. Nice literature review and introduction
4. Experimental comparison with various existing methods
5. In my view, the proposed $\ell_{\infty}$ coreset for $\ell_1$ regression in Lemma 2.7 is more interesting and solid than the main theorem and coreset design for neural networks.
6. Details of the coreset design with complexity analysis of algorithm

**Main Concern**
1. The proposed bound strongly depends on the regression complexity measure, which is not proven to be **bounded**. Notably, we can design sets for which this complexity measure scales with the size of the set. For such a set, the established coreset-size is greater than the number of neurons, hence the theoretical bound is vacuous.
2. To compare the baselines, I recommend checking comparisons for networks with the same accuracy levels and comparison rates (in tables 1-3). Indeed, the compression has to be conducted on the same input networks (so with the same level of accuracy). Comparing compression algorithms on different networks may not lead to a fair comparison.


**Minor Comments**
 Adding details about the computational complexity of methods to find Lowner ellipsoid would improve clarity.

**Post Rebuttal**
The response address my major concerns so I updated my score.

---

> ### Author Response · Authors · 2022-08-02
> **Response to Reviewer 4TWt: Part 1. Opening statement, and answering the first 2 questions**
>
> **We thank the reviewer for their time and effort in reviewing our paper and greatly appreciate the questions raised and the constructive criticism of our work. The insightful comments are very much appreciated.**
>
> We first answer the reviewer questions:
>
> **Q1:** There are coreset designed for l1 regression with similar techniques:(i) Lowner ellipsoid and (ii) Caratheodory set (see, for example, the survey on Sampling Algorithms and Coresets for  Regression). I wondered what the novelty of the proposed algorithm compared to these exciting methods for l1 regression.
>
> **Answer:** We thank the reviewer for raising this. The novelty in our approach is that it can be extended to work with every $\ell_p$-regression as it aims to be independent of the problem at hand, which most coresets in this field don't guarantee. The running time is also better than most coresets. Indeed, with the use of matrix sketches, our methods can even take less time at the expense of increasing the probability of failure.
>
> Furthermore, the observations in the context of activation functions and model compression (as stated in the submitted version in Section 2.2) are indeed novel.  The novelty of our approach in this context exploits the following observations.
>
> - (i) Since activation functions $\phi$ are usually continuous non-decreasing,  for every query $x$ and a set  $P$, the largest contribution to $\sum_{p\in P} \phi(p^Tx)$ is associated to a point on the convex hull of $P$. Hence, we find geometrical body $B$ of a bounded number of vertices, that is enclosed in the convex hull of ${P}$ and with some expansion, $B$ encloses the convex hull of ${P}$. Now, each point on the boundary of the convex hull of $P$ can be represented as a convex combination of two points $p_1,p_2$, where $p_1$ on $B$ and  $p_2$ on its dilated form. We leverage, L\"{o}wner ellipsoid for this purpose, however, the observation is novel.
>
> - (ii)  Now, we represent $p_1$ and $p_2$  by points from $P$.  $p_1$ can be represented via a convex combination of $d+1$ points from $P$. The same holds for  $p_2$ but via a conical combination (linear combination where the weights are non-negative). We now invoke Carath\'{e}odory theorem for this purpose, however, the observation is novel.
>
> - (iii) Finally, since some functions, including the ReLU function, do not admit an $\varepsilon$-coreset of size $o(n)$ as shown in Mussai et.al. Thus, we use a generalized form of what is known as the complexity measure of a set of points, which was first introduced in Munteanu et.al.  The usage of this measure in such a context is novel. We use this measure to determine the complexity of a given set $P$ with respect to ReLU, and the coreset size theoretically.
>
>
> ----
>
> **Q:2** Can authors validate the boundedness of regression complexity measures for simple cases? Is it possible to characterize a worse case for the regression complexity measure? |  The proposed bound strongly depends on the regression complexity measure, which is not proven to be bounded. Notably, we can design sets for which this complexity measure scales with the size of the set. For such a set, the established coreset-size is greater than the number of neurons, hence the theoretical bound is vacuous.
>
> **Answer**The reviewer raised a very good point. There exists an example where the complexity measure is unbounded. To imagine such a case, consider a set of points distributed evenly on a unit ball. In this case, you can always find a hyperplane separating one point from the rest, leading to an infinite complexity measure. This is in the context of having Relu as the loss function. This case is not surprising since also in the context of coresets, for such datasets, the lower bound on the coreset size is the size of the entire dataset as was proven in Mussay et. al. Note that such cases are not the norm, but rather end cases used to show the problems raised by the characteristics of the Relu loss function.
>
> Theoretically, the complexity measure boils down eventually to how free can the bias term be (the last entry of x). This term is the only thing that can ensure that one point can be separated from the rest in the sense of finding a seperating hyperplane, leading to an infinite complexity measure. So in a sense, if we would set bounds on this term, our complexity measure from a theoretical point of view is bounded.
>
> Our experimental results show the most used datasets don't admit such behavior. As for bounding the complexity measure for the case of the datasets, we can use a variant of the algorithm described in the proof of Theorem 3 of "On Coresets for Logistic Regression".by Munteanu et. al.
>
> Finally, we note that the complexity measure is not new to the world of coresets. Specifically, it was first defined in Muntenau et. al. and further used in many other coreset papers. To our knowledge, we were the first to use it in the context of model compression.

---

> > ### Comment · Reviewer_4TWt · 2022-08-04
> > **Clarification about run time**
> >
> > Would you please explain your algorithm is faster in practice or its computational complexity is provably lower? You are comparing with which baseline algorithm?

---

> > > ### Author Response · Authors · 2022-08-05
> > > **Run time of algorithm with comparison with previous related works**
> > >
> > > With pleasure. The time complexity in our paper is indeed faster than the algorithm proposed in Dasgupta et. al. As for the paper of Mai et. al. while our proposed method is slower, our algorithm is more applicable to other loss functions including the ones supported by Mai et. al.
> > >
> > > In addition, the coreset size proposed in Mai et. al. is quadratic in the complexity measure and linear in the dimension. Our approach on the other hand is polynomial in the rank of the points (smaller than the dimension when $r < d$), and linear in the complexity measure. When the data at hand imposes a complexity measure that is of polynomial order in the dimension or the rank, our method while being slower than that of Mai et. al. attains a smaller coreset with the same desired approximation factor.

---

> > ### Comment · Reviewer_4TWt · 2022-08-04
> > **More details about the complexity bound**
> >
> > Since you noted "The usage of this measure in such a context is novel". The complexity measure needs to be studied and justified your context. In particular, you need to elaborate examples where the complexity measure is lower than the number of neurons.

---

> > > ### Author Response · Authors · 2022-08-05
> > > **Complexity bound - clarification**
> > >
> > > We would like to express our deep gratitude for the concerns raised by the reviewer, and we hope that the added information will contribute to a better understanding of our paper.
> > >
> > > First, we note that while the complexity measure was first defined for constructing coresets with respect to the logistic regression problem, it also has been used for the ReLU regression problem (minimizing the sum of ReLU losses) by Mai et. al. Throughout these papers, the authors expected the complexity measure to be small other than in some cases. In our paper, we also operate under the same assumption, i.e., the complexity measure is reasonably small, where the input data in our experimental setting is the model's weights.
> > >
> > > With this in mind, we hope that the following examples will shed light on the boundness of this term:
> > >
> > > For simplicity, given a set $P$ (expressing our neurons) containing points in $\mathbb{R}^3$, where the points in $P$ lie on a 2-dimensional affine subspace parallel to the $xy$-plane.  Notice that in our setting, the complexity measure is defined as the maximal value of an optimization problem involving $P$ and a set of queries $X:=\mathbb{R}^2 \times \left\lbrace 1 \right\rbrace$.
> > >
> > > If we were able to find a hyperplane whose normal in $X$, such that one point from $P$ can be separated from the rest of the points in $P$, then the complexity measure can reach $\infty$. Such an example is given through the following link: https://postimg.cc/8jjkj8X8
> > >
> > > As seen in the image (referenced through the website link), a clear separation can be made between one point and the rest, leading to two sets of points: The first set contains one single point that has a positive dot product with the normal $x$ to the separating hyperplane, while the other set contains the remaining points, each with negative dot product with $x$. This leads to a large complexity measure, and as the separating hyperplane gets closer to the set containing the single point, the complexity measure increases, i.e., the closer it is, the higher the complexity measure, as it can tend to infinity.
> > >
> > > On the other hand, when one can not separate a single point or a minimal set of points from the rest of the data, we expect the complexity measure to be small. To illustrate such an example, please refer to: https://postimg.cc/vDmbNFs1
> > >
> > > In such an example, one will find that the input data is centered around the origin, thus, such data is not linearly separable leading to small complexity measures. This is due to the fact that the $x$ represents the normal to a hyperplane emerging from the origin.
> > >
> > > In fact, during our experiments, we noticed that the complexity measure in the case of LeNet300-100 was around 15 when the input data (matrix representing the neurons) had 300 rows (points). Observe that, it is common in coreset literature that practically you would use sample sizes that are smaller than the bound on the coreset size since such bounds are pessimistic in nature. Hence, during our experiments, we didn't incorporate the complexity measure in our sample size or in our computation of the sensitivity. The reason behind not using the complexity measure in our sensitivity is due to the observation that such a term will be eliminated when computing the sampling probability (see Theorem B.2 in the appendix), i.e., it will have no effect on the sampling procedure other than the coreset size (from a theoretical point of view).
> > >
> > > Our experiments confirm such an observation, i.e., we obtain coresets with favorable results even when the complexity measure is not incorporated in our computations, or when the chosen sample size is smaller than the bound on the coreset size.
> > >
> > > The appendix in the supplementary material has been modified in light of this.

---

> ### Author Response · Authors · 2022-08-02
> **Response to Reviewer 4TWt: Part 2. Adressing the remaining comments**
>
> We now answer the remaining comments:
>
> **Comment 1:** To compare the baselines, I recommend checking comparisons for networks with the same accuracy levels and comparison rates (in tables 1-3). Indeed, the compression has to be conducted on the same input networks (so with the same level of accuracy). Comparing compression algorithms on different networks may not lead to a fair comparison.
>
> **Answer:** We first note that the tested architectures/models are indeed the same as our competing methods. However, the baseline model error may differ between one method and the other, because we train the network first - a training process usually includes randomness in it. Hence the input network parameters may be different but the model architecture is the same. Indeed this is known and common in the context of pruning.
> As for the compression rates,
>
> - (i) In all of the tested networks, we tried to report the highest compression rate, or at least to be very close to it, while achieving better or comparable results with all baselines. Hence, we think that having a higher compression rate and the best accuracy (or very close to it) across all networks shows the robustness of our approach.
>
> - (ii) In the context of pruning it is common to compare results by tables, where each reported method has one compression rate
>
> ----
>
> **Minor Comments:** Adding details about the computational complexity of methods to find Lowner ellipsoid would improve clarity.
>
> **Answer:** We thank the reviewer for this suggestion. It has been incorporated.
>
> ----
>
> **Again, thank you for the clear and detailed comments.**

---

### Official Review · Reviewer_B4JF · 2022-07-11

**Rating:** 7
**Confidence:** 3
**Soundness:** 3 good
**Presentation:** 3 good
**Contribution:** 3 good

**Summary:**

This paper aims to advance coreset-based neural network pruning by relaxing the assumptions made in the previous works (especially Mussay et al.). This work claims to provide data-independent sample complexity bounds for layer pruning for multiplicative coresets including for ReLU-like activations which were considered to be non-approximable with fewer than O(n) samples. Novelty aspects come by providing a different upper bound for sensitivity, to be used for importance sampling. This upper bound on sensitivity uses certain conditions satisfied in Algorithm 1 which uses the Carathéodory theorem applied to the vertices of maximal volume ellipsoid contained within the convex hull of weights in a layer. Having mentioned these theoretical results, the paper empirically shows the efficacy of their algorithm on different architectures (LeNet, VGG, and ResNet) on different datasets (MNIST, CIFAR-10, and ImageNet-1K).

**Questions:**

As mentioned in the main review.

**Limitations:**

As mentioned in the main review.

**Strengths And Weaknesses:**

Strengths:

- The theory seems to be reasonably sound and novel (usage of Löwner-John ellipsoid with Carathéodory theorem).
- The Paper is decently written albeit I’ve some questions (described later)
-  Empirical experimentation involves a variety of networks and datasets and baselines.

Weaknesses:

Weaknesses of this work are mostly my questions and some suggestions which call for small experimentation for coreset guarantee. I describe them as follows:

- Since the work follows the same notations as in Mussay et al. can they also include a neural network diagram so that it would be easier to follow what is mentioned in line 77?
- Line 699 claim 1: why does the ellipsoid have at most 2r vertices? Shouldn't that be exactly 2r?
- What is special about the vertices of the ellipsoid that is used to get the Carathéodory set?
- The upper bound on sensitivity (and hence on the normalizing constant t) is in terms of regression complexity measure $\mu(P)$. Is there any bound on this term, or this can get arbitrarily large (somewhere as large as the o(n)?).
- I am unable to comprehend why following this procedure would lead to the possibility of multiplicative coresets. The proof given by Mussay et al. (Appendix A.1) describes a possibility of the case that shows that for ReLU like activation functions sample complexity is O(n). Why should this procedure not fall into the case mentioned by Mussay et al.?
- In Definition 2.5, it may be better to use $x \in \mathbb{R}^{d} \times {1}$ and $P \subseteq  \mathbb{R}^{d+1} $.
- Algorithm 2:  r is not defined.
- Algorithm 2: To run the while loop runs iff $|Q| \geq 2*rank(Q)^2$. Where did this condition come from? In neural network pruning setting where |Q| at the beginning would be the #neurons in the next layer and rank(Q) can be at max #neurons in the next layer, how often does this while loop ever get executed?
- On the empirical side: similar to Mussay et al, can authors include a curve showing true multiplicative error in some simulations, comparing it with baselines, showing the efficacy of their method?
- Can authors include pruning results without re-training as well like Mussay et al.? It is important to see how performance looks when not re-trained.
- All the results need to have the standard deviation attached for the testing error.
- Table 1: Why is the PR different for the methods but PvC or  Algo2?


Add the following to the related work:

- Gantavya Bhatt and Jeff Bilmes. Tighter m-DPP Coreset Sample Complexity Bounds. In ICML 2021 Workshop: SubSetML: Subset Selection in Machine Learning: From Theory to Practice, Virtual, July 2021. This paper aims to improve bounds for DPP-based coreset and show possible applications to neural networks pruning (similar to mussay et al.)

- Diversity Networks: Neural Network Compression Using Determinantal Point Processes. Zelda Mariet, Suvrit Sra


I'm willing to raise my rating upon satisfactory response.

---

> ### Author Response · Authors · 2022-08-02
> **Response to Reviewer B4JF: Part 1. Opening statement, and answering the first 5 questions**
>
> **We would like to thank the reviewer for the positive evaluation of our work, professional reviewer, and highly knowledgeable comments to help improve our manuscript.  We also appreciate the reviewer’s willingness to change the score if their concerns are sufficiently addressed. This is indeed motivating.**
>
> We now answer the reviewer's questions.
>
> **Q1:** Since the work follows the same notations as in Mussay et al. can they also include a neural network diagram so that it would be easier to follow what is mentioned in line 77?
>
> **Answer:** We thank the reviewer for this suggestion as it indeed makes the idea clearer. We note that, in Line 88, we referred the reader to Figure 1 in Mussay et al. The main reason was to avoid duplications. However, following your comment, we are working on adding a diagram of our own which will clarify the details. A new revision will be submitted very soon, including these changes.
>
> ----
>
> **Q2:** Line 699 claim 1: why does the ellipsoid have at most 2r vertices? Shouldn't that be exactly 2r?
>
> **Answer:** You are right. It has been fixed. Thanks for the careful reading.
>
> ----
>
> **Q3:** What is special about the vertices of the ellipsoid that is used to get the Carathéodory set?
>
> **Answer:** We have shown that the vertices can be used in order to describe every point on the convex hull of the input data (in our end, it would the network's weights). If we can describe these points using the Carathéodory set, we can ultimately represent every point on the convex hull of the input data. Following your comment, we clarified the importance of the aforementioned vertices to our coreset design.
>
> ----
>
> **Q4+Q5:** The upper bound on sensitivity (and hence on the normalizing constant t) is in terms of regression complexity measure. Is there any bound on this term, or this can get arbitrarily large (somewhere as large as the o(n)?) | I am unable to comprehend why following this procedure would lead to the possibility of multiplicative coresets. The proof given by Mussay et al. (Appendix A.1) describes a possibility of the case that shows that for ReLU-like activation functions sample complexity is O(n). Why should this procedure not fall into the case mentioned by Mussay et al.?
>
> **Answer:** Thank you for the reasonable questions. The answer is related to the complexity measure that we used and the input data (the network in our case). First, not all datasets require sample complexity of O(n) when using ReLU as an activation function. Such a result is backed by our experimental section.
> Indeed, there exists an example where the complexity measure is unbounded. To imagine such a case, consider a set of points distributed evenly on a unit ball. In this case, you can always find a hyperplane separating one point from the rest, leading to an infinite complexity measure. This is in the context of having Relu as the loss function. This case is not surprising since also in the context of coresets, for such datasets, the lower bound on the coreset size is the size of the entire dataset as was proven in Mussay et. al. Note that such cases are not the norm, but rather end cases used to show the problems raised by the characteristics of the Relu loss function.
>
> Theoretically, the complexity measure is influenced by how free can the bias term be (the last entry of x). This term is the only thing that can ensure that one point can be separated from the rest in the sense of finding a separating hyperplane, leading to an infinite complexity measure. So in a sense, if we would set bounds on this term, our complexity measure from a theoretical point of view is bounded. From a practical point of view, one need not compute such a term (or in other words, assume any assumption on the bias), and as our experiments indicate, we still lead to very good results.

---

> > ### Author Response · Authors · 2022-08-02
> > **Response to Reviewer B4JF: Part 1. Opening statement, and answering the first 5 questions - Continued**
> >
> > In the context of model pruning, from the perspective of the complexity measure, the model's weights are the input denoted by a matrix $P \in \mathbb{R}^{n \times (d+1)}$, while the query is now $\mathbb{R}^d \times \left\lbrace 1 \right\rbrace$. Thus the complexity measure is now $\mu\left( P \right) := \sup_{x \in \mathbb{R}^d \times \left\lbrace 1 \right\rbrace} \frac{\sum_{q \in \left\lbrace p \in P \middle| p^Tx\leq 0 \right\rbrace} -\left| q^Tx \right|}{\sum_{q \in \left\lbrace p \in P \middle| p^Tx > 0 \right\rbrace} \left| q^Tx \right|}$. With this in mind, we observe that the complexity measure is now an instance of the complexity measure used in Mai et. al. The complexity measure now relies entirely on the structure of the model's weights, where the goal is to find the largest ratio between the sum of the absolute of the values inside the rectified neurons prior to applying the rectification, and the sum values of non-rectified neurons. As for bounding this complexity measure, we can use a variant of the algorithm described in the proof of Theorem 3 in Munteanu et. al.

---

> ### Author Response · Authors · 2022-08-02
> **Response to Reviewer B4JF: Part 2. Answering the remaining questions**
>
> **Q6:** In Definition 2.5, it may be better to use $x\in R^{d} \times 1$, and $P\subset R^{d}$
>
> **Answer:** We thank the reviewer for this suggestion. It has been incorporated.
>
> ----
>
> **Q7:** Algorithm 2: r is not defined.
>
> **Answer:** Fixed. Thanks for the careful reading. $r$ is indeed the rank of $Q$.
>
> ----
>
> **Q8:** Algorithm 2: To run the while loop runs iff Q>2rank(Q)^2 . Where did this condition come from? In neural network pruning setting where |Q| at the beginning would be the #neurons in the next layer and rank(Q) can be at max #neurons in the next layer, how often does this while loop ever get executed?
>
> **Answer:** This condition rises from the fact that the l-infinity coreset must contain up to $2rank(Q)^2$ points. In other words, if the size of $Q$ is less than that term, we halt the algorithm from constructing l-infinity coresets and simply use all of $Q$ as the final l-infinity coreset. In the literature on leveraging the use of coresets for the task of model compression, one problem that always arises is the dimensionality of points, where usually the dimension is much higher than or equal to the number of points, and unfortunately, the VC-dimension usually depends on the dimension of the data. Such a case usually leads to the observation that the rank of $Q$ is simply the number of points. For example, in previous works (like Mussay et.al.), the coreset size always contained the number of neurons of the previous layer, this arises from the VC dimension of the problem at hand. As the reviewer raised such an important concern, and indeed it is concerning since in this case, our algorithm basically states that uniform sampling is used.
>
> This is where the dimensionality reduction comes, to ensure that the rank of Q would be much less than the square root of the number of neurons. The dimensionality reduction leads to small coresets and still attain favorable results. We hope that our work will inspire future generations of papers investigating whether it is possible to maneuver around this problem leading to coresets with less dependency on the dimension of the model while not imposing restrictive constraints which were used in previous works.
>
> ----
>
> **Q9 + Q10:** On the empirical side: similar to Mussay et al, can authors include a curve showing true multiplicative error in some simulations, comparing it with baselines, showing the efficacy of their method? |  Can authors include pruning results without re-training as well like Mussay et al.? It is important to see how performance looks when not re-trained.
>
> **Answer:** We thank the reviewer for both suggestions. We are now working on adding these graphs/tables, hopefully, we will add them very soon.
>
>
> ----
>
> Q11: All the results need to have the standard deviation attached for the testing error.
>
> **Answer:**Added.
>
> ----
>
> Q12: Table 1: Why is the PR different for the methods but PvC or Algo2?
>
> **Answer:** In all of the tested networks, we tried to report the highest compression rate across all methods, or at least to be very close to it, while achieving better or comparable results with all baselines. Hence, we think that having a higher compression rate and the best accuracy (or very close to it) across all networks shows the robustness of our approach.
> We note that we are adding more results on different compression rates.
>
> ----
>
> **Finally, we have indeed added the suggested citations. Thank you for the professional review.**

---

> > ### Author Response · Authors · 2022-08-04
> > **Additional experiments**
> >
> > Following the fruitful suggestions raised by the reviewer on highlighting the abilities of our coreset via additional experiments, the main paper has been modified. The experiments can be found in the supplementary file, which contains the main paper with the appendix.

---

> > ### Comment · Reviewer_B4JF · 2022-08-06
> > **Thanks for the response!**
> >
> > Most of my concerns are satisfactorily addressed, and I've raised my rating.

---

### Official Review · Reviewer_Uwaw · 2022-07-11

**Rating:** 7
**Confidence:** 3
**Soundness:** 3 good
**Presentation:** 3 good
**Contribution:** 3 good

**Summary:**

This paper proposes a method for pruning neural network weights at each layer. This improves on previous algorithms for activation function coresets by attaining tight sensitivity bounds with no data assumptions and mild assumptions on the network weights. The coreset construction relies on combining two key ideas from convex geometry: specifically finding the Caratheodory set of each vertex of a John-Löwner ellipsoid. Experiments show that the proposed approach outperforms several recent baselines, pruning more weights while achieving lower classification error.

**Questions:**

- Why different baselines used for different tables/models/experiments?
- Is Theorem 2.8 directly comparable with previous work?
- Do you have any intuition/empirical results on what dimensionality reduction method works best in this setting?
- Can this method be applied to attention-based networks such as transformers?


**Limitations:**

This paper describes limitations as opportunities for future work, but does not address any potential negative social impact.

**Strengths And Weaknesses:**

## Originality/Significance/Quality
- Builds on previous work on network pruning in a novel way via tools from convex geometry
- Cites related work where appropriate
- Elegant idea that with intermediate results that may be of independent interest
- Experimental results look impressive, although different baselines are used in different experiments

## Clarity
- The paper is generally a pleasure to read, but I have a couple of suggestions:
    - The weights assumption (Regression Complexity Measure) is a key contribution of the paper, and the presentation would be improved it if was mentioned earlier, perhaps with an informal statement and/or background motivation in Section 1
    - The John ellipsoid is defined with a dilation of $d$ (ambient dimension), but is used in Lemma 2.7 with a dilation of $r$ (rank).
- Typos
	- Line 140: "with respect ReLU" should be "with respect to ReLU"
	- Line 148: "at [74]" should be "in [74]"
	- References [5-7] are identical

---

> ### Author Response · Authors · 2022-08-02
> **Response to Reviewer Uwaw: Part 1. Opening statement, and addressing the reviewers comments**
>
> **We thank the reviewer for the positive feedback, professional review, and constructive comments. We greatly appreciate the time and effort of the reviewer to help us improve our work.**
>
> We first address the reasonable comments of the reviewer.
>
> **Comment 1:** The weights assumption (Regression Complexity Measure) is a key contribution of the paper, and the presentation would be improved it if was mentioned earlier, perhaps with an informal statement and/or background motivation in Section 1
>
> **Answer:** This is a very important comment. Indeed, the complexity measure reflects upon the hardness of computing coresets in this field for the sake of model compression. We showed that in order to compute an epsilon coreset, its size depends on this measure.
> Following this comment, we have updated the presentation and mentioned it earlier.
>
> ----
>
> **Comment 2:** The John ellipsoid is defined with dilation of (ambient dimension), but is used in Lemma 2.7 with dilation of (rank).
>
> **Answer:** Thank you for the very careful reading. First, we note that the John ellipsoid definition can be modified to take into count the rank of the input points and not the dimension. That is why we used it this way in Lemma 2.7. Note that the Lowner ellipsoid is computed on points from the $\mathbb{R}^r$ space, specifically $P^\prime$ (see definition of $P^\prime$ at Line 2 of Algorithm 1), where $r$ denotes the rank of the subspace represented by $Y$.
>
> ----
>
> **Comment 3:** Line 140: "with respect ReLU" should be "with respect to ReLU" | Line 148: "at [74]" should be "in [74]" | References [5-7] are identical
>
> **Answer:** Fixed. Thanks for the careful reading.
>
> Note that reference [7] was not the same as [5] and [6], so we kept it as it is. Reference [5] was removed.

---

> > ### Comment · Reviewer_Uwaw · 2022-08-08
> > **Follow up**
> >
> > Thank you for removing the duplicate reference and for answering my questions about baselines, clarity, and dimensionality reduction.

---

> ### Author Response · Authors · 2022-08-02
> **Response to Reviewer Uwaw: Part 2. Answering the reviewers questions**
>
> We now address the questions raised by the reviewer.
>
> **Q1:** Why are different baselines used for different tables/models/experiments?
>
> **Answer:** The reviewer raises a good point. The reason for this is that the experimental results of the competing methods did not span across all the datasets and networks which were tested with our technique. So as common in the context of network pruning, for each network, we report the relevant competing methods which have conducted experiments on the same network.
>
> ----
>
> **Q2:** Is Theorem 2.8 directly comparable with previous work?
>
> **Answer:** Yes.
> - 1.  Our coreset supports different activation functions without the need to change the sensitivity that much. Specifically, it will only be multiplied by some scalar, unlike previous coresets where different losses impose drastically different sensitivities/leverage scores and algorithms. This is since our coreset unlike other coresets is in its essence a framework of coresets for different $l_p$ losses, as it can be used as is for different losses and yet still attain epsilon approximation. Unlike our approach, previous methods required different computations/algorithms when the $p$ in $l_p$ regression changes.
>
> - 2. when $r$ is small, then our method outperforms previous methods, while, when the data is of full rank, then previous methods for a specific cost function may be better.
>
> ----
>
> **Q3:** Do you have any intuition/empirical results on what dimensionality reduction method works best in this setting?
>
> **Answer:** In our setting, we saw that the JL lemma is a good candidate for dimensionality reduction. We further note that the main downside of using JL is that it provides bounds on the dimensions of the lower dimensional space required to attain provable approximation. On the other hand, TSNE or UMAP (for instance) which support non-linearity, do preserve the structure of the data by "preserving the relationship between every pair of points". While JL Lemma proved to be useful in our experimental setting, we believe that TSNE/UMAP will yield better results due to their advantage in capturing much more complex information than any linear-based dimensionality reduction method.
>
> ----
>
> **Q4:** Can this method be applied to attention-based networks such as transformers?
>
> **Answer:** We believe it can be extended to such models, but some work still needs to be done. We first need to define the query space for other layers and techniques in transformers. Some of these layers are easily supported via our current approach such as embedding layers that can be viewed simply as matrix multiplications. Other layers still need to be mathematically formulated in the context of coresets, such as the attention layer.  As this raised question is indeed very interesting, we leave this as a future work where our method can be further extended to fully support such beasts of deep neural models.
>
> **Finally, thank you for the professional review.**

---

### Official Review · Reviewer_8WWu · 2022-07-11

**Rating:** 5
**Confidence:** 4
**Soundness:** 2 fair
**Presentation:** 2 fair
**Contribution:** 2 fair

**Summary:**

This paper presents a new method for network pruning based on coresets which relies on inner and outer approximations of the convex hull of the weights of each layer of the neural network, with the inner approximation being obtained from a subset of the weights of the neurons and the outer approximation being a dilation of the former. This form of determining which weights to prune is almost completely independent from the dataset, which is one of the main advantages claimed by the authors.

**Questions:**

1) In Section 1.1, line 42, what do you mean by finding a query x*? My understanding is that you are actually looking for a subset of all queries, not for a single one.

2) In Section 2, line 83, what do you mean by w(p_i) := w_i? I understand that every neuron of the top layer has a weight for the neuron at the bottom layer, but it is not a function of the value of its weights.

3) What is the reason for assuming all the weights of P as equal to 1 in the main paper?

4) In Section 2.1, between lines 114 and 116, what do you mean by "Once we bound these sensitivities" and "The size of the sample is proportional to the sum of these bounds"?

5) In Theorem 2.3, line 127, what do you mean by "points with nonempty interior"? Points do not have interior.

6) In Section 2.2, line 133, can you explain what do you mean by the dilated form of B consisting of a conical combination? Wouldn't it make more sense to say that this is a weighted sum in which the sum of the weights now exceed 1?

7) In Section 3, why is every network pruned with a different subset of approaches? What happened with the missing ones in each of those cases?

8) In Table 1, what is the meaning of "Baseline Err." and "Pruned Err."? Is the first the complement of the accuracy and the second the drop due to pruning? Why is the first one almost the same but not always across all methods tested?

9) In Table 2, what is the meaning of "FR"?

10) In Table 3, is "Orig. Err." the same as "Baseline Err." from before? If so, please make it uniform.

11) Why is only one pruning rate reported for each model and approach?

12) How do you think coreset-based pruning relates to what happens in the case exact compression methods (e.g., https://arxiv.org/abs/2102.07804 & https://arxiv.org/abs/2107.07467)?

**Limitations:**

No; Section 5 talks about future work, which is not the same thing as limitations of the current work.

**Strengths And Weaknesses:**

This paper presents some interesting ideas, but it is not easy to follow. I have several comments to the authors about how to improve the writing below. Among other things, some concepts are used before definition, some explanations are repeated, and often it is only possible to understand what is going on when a new and more accessible definition is presented. However, what really concerns me is the limited experimental evaluation: the last sentence of the abstract sounds an alarm of hand-picking. Indeed, the results in Section 3 are very limited: only one pruning rate is used for each method tested, which is far from ideal. I would not expect a new method to always beat all other methods in order to accept a paper, but I do expect to see a thorough assessment of where and why a new method performs better. Finally, I believe that the authors understand quite well what they are doing and are very excited about it, but they also need to provide more context for the readers by going over their design choices in more detail and with more context and explanations to non-experts.

### Comments on the writing

Section 1:
- In the Abstract, line 5, you write data dependant. In the Introduction, line 61, you write data-dependent. Please choose one convention for spelling and hyphenation.
- The whole paragraph on coresets is very difficult to read and I could only understand most of it (except for my question about x*) after reading the paragraph "How to prune using coresets?" in Section 2, in which the useful parts are actually repeated. I would encourage the authors to rewrite this and perhaps merge both paragraphs at a convenient point.
- Line 46: "corestes" -> coresets
- Line 69: "guarantees" -> guarantee

Section 2:
- Line 88: "See" -> see
- By the time that [n] is defined in line 92, it has already been used in line 83.
- Definition 2.2 repeats a lot of terms that were already introduced in the second paragraph of Section 2. Again, I would encourage the authors to think of rewriting the paper with less repetition, more clarity, and each term introduced at the most appropriate time.
- Line 113-114: the excerpt "and it corresponds to the importance of this point with respect to the query space at hand" seems to refer to s(p) rather than its denominator alone, but it is not clear from context.
- Line 118: "Our method hinges upon an elegant combination of": Academic papers should not contain self-praise; please remove "elegant". Likewise for "an elegant combination" in line 326.
- Line 137: One of the many examples of the use of "e.g." looks a bit unusual: are you saying that ReLU, among many others, does not admit such an approximation? Please also revisit other uses in the paper (line 148) and compare them with the form in which this abbreviation is commonly used.
- Line 139: Replace "at" with "in" (two incidences; see also lines 148, 156, 194 twice, 195, 208, 286)
- Caption of Figure 1: Replace "novelty" with "contribution"? In the fourth line, replace "A" with "a"
- Line 161: Add "an" after "constructing"
- Line 221: "Deep" -> deep

Section 3:
- Line 254: Remove quotes from "Numpy"
- Line 293: "obtain a" -> obtain an; remove comma between "coreset" and "when"

Section 4:
- Line 322: "Decision Trees" -> decision trees

Section 5:
- Line 328: "on variety" -> on *a* variety

References:

- Remove [5]: duplicate of [6] with wrong publication year

---

> ### Author Response · Authors · 2022-08-02
> **Response to Reviewer 8WWu: Part 1. Opening statement, and addressing the "comments on the writing"**
>
> ***We thank the reviewer for the professional review, careful reading, and clear detailed comments. Your insightful review has undoubtedly aided us in improving our paper.***
>
>
> First, we note that all of the "Writing comments" have been addressed and fixed, thanks to your careful reading. Here are the detailed changes.
>
> Section 1
> --------
>
>
> **Comment 1**: data-dependent or data dependent. Please choose one convention for spelling and hyphenation.
>
> - **Answer**: We replaced every occurrence of "data dependant" with "data-dependent".
>
> ----
>
> **Comment 2**: the whole paragraph on coresets is very difficult | perhaps merge with the paragraph "How to prune using coresets".
>
> - **Answer**: We have merged the paragraph on coresets with the one on "How to prune with coresets." We reworked these paragraphs into a single paragraph that contains all of the pertinent information clearly.
> We will publish/share the revised version in the coming days, and we hope you will find it satisfactory.
>
> ----
>
> **Comment 3**: Line 46: "corestes" -> coresets | Line 69: "guarantees" -> guarantee
>
> - **Answer**: All typos were fixed.
>
> ----
>
> Section 2
> --------
>
> **Comment 4**: Line 88: "See" -> see |  Line 139: Replace "at" with "in" (two incidences; see also lines 148, 156, 194 twice, 195, 208, 286) |Caption of Figure 1: In the fourth line, replace "A" with "a" | Line 161: Add "an" after "constructing" |Line 221: "Deep" -> deep
>
> - **Answer**. Done. Thanks for the careful reading.
>
> ----
>
> **Comment 5**: [n] is defined in line 92, it has already been used in line 83
>
> - **Answer**: Fixed. We replaced [n] by {1,..,n} at line 83. Thanks for noticing and commenting.
>
> ----
>
> **Comment 6**: Definition 2.2 repeats a lot of terms that were already introduced in the second paragraph of Section 2. Again, I would encourage the authors to think of rewriting the paper with less repetition, more clarity, and each term introduced at the most appropriate time.
>
> - **Answer**: Following the earlier suggestion (Comment 2) on integrating the two paragraphs, we have adjusted the text such that Definitions 2.1 and 2.2 are presented immediately with these paragraphs -> to make it clearer for the reader and to avoid duplications.
> Furthermore, following your feedback, the entire document had considerable writing revisions, not just in the parts you commented on. Again, the revised version will be submitted very soon.
>
> ----
>
> **Comment 7**: the excerpt "and it corresponds to the importance of this point with respect to the query space at hand" seems to refer to s(p) rather than its denominator alone, but it is not clear from the context.
>
> - **Answer**: You are right, it refers to s(p). We have rephrased this sentence, so it will be correct. Thanks for pointing this out.
>
> ----
>
> **Comment 8**: remove "elegant".
>
> - **Answer**: Removed.
>
> ----
>
> **Comment 9**: Line 137: One of the many examples of the use of "e.g." looks a bit unusual: are you saying that ReLU, among many others, does not admit such an approximation? Please also revisit other uses in the paper (line 148) and compare them with the form in which this abbreviation is commonly used.
>
> - **Answer**: To make it clear, we have rephrased these sentences as follows:
> The sentence in Line 137: “Finally, it is known that some functions, including the ReLU function, do not admit an $\varepsilon$-coreset of size $o(n)$”.
> The sentence in Line 148: “our results can be easily extended to a wide family of activation functions including the Sigmoid function”.
> Thanks for pointing this out.
>
> ----
>
> **Comment 10**: Caption of Figure 1: Replace "novelty" with "contribution"?
>
> - **Answer**: In this figure, we illustrate the main novel ideas behind our technique, and how we utilize a combination of tools from computational geometry and convex analysis to prune neural networks (via sensitivity sampling). We believe that the word “novelty” fits this description, however, indeed, if the reviewer believes that the word “contribution” describes the figure better, we will be very happy to adapt.
>
> ----------------------------------------------------------------------------------------------------
>
> Section 3, Section 4, Section 5, and References.
> -----------------------------------
>
> **Comment 11**:  Line 254: Remove quotes from "Numpy" |Line 293: "obtain a" -> obtain an; remove comma between "coreset" and "when" |Line 322: "Decision Trees" -> decision trees |Line 328: "on variety" -> on a variety | Remove [5]: duplicate of [6] with wrong publication year
>
> - **Answer**: All of the detailed clear comments on these sections have been addressed and fixed. Thanks.

---

> ### Author Response · Authors · 2022-08-02
> **Response to Reviewer 8WWu: Part 2.  Answering the first 4 questions raised by the reviewer**
>
> We now address the questions raised by the reviewer.
>
> **Q1:** In Section 1.1, line 42, what do you mean by finding a query x*? My understanding is that you are actually looking for a subset of all queries, not for a single one.
>
> **Answer:** This statement tries to explain that in optimization problems (or machine learning in general), the goal is usually to find a query that minimizes (or maximizes) some cost function. As for your question, it is easy to forge a link between "coresets" to "solving an optimization problem". This is done as follows.
>
> - In the context of coresets, the goal is to find a small weighted subset such that for a given cost function, the cost of applying any solution (hypotheses/query) on the coreset approximates the cost of applying the same solution on the whole data.
> With this being said, since a coreset approximates the cost of every query, we do note that in many cases, coresets are applied for approximating the optimal solution. Specifically, solving the desired optimization problem on the whole data can be a hard problem when the time needed for such a solution is either polynomial or exponential in the size of the whole data, or when the required memory is too high. In this case, coresets can be leveraged such that the optimal solution of fitting the coreset can provably result in $(1+4\varepsilon)$-approximation towards the optimal cost of solving the optimization on the whole data (the proof is very easy, it is done by applying the triangle inequality few times). In other words, we can solve the problem on the coreset to obtain a solution x*, and then apply x* to the whole data giving a good approximation for solving the problem from the beginning on the whole data. These details have been added to the appendix.
> ----
>
> **Q2:** In Section 2, line 83, what do you mean by w(p_i) := w_i? I understand that every neuron of the top layer has a weight for the neuron at the bottom layer, but it is not a function of the value of its weights.
>
> **Answer:** You are right. Every neuron in the top layer has a weight for the $i$th neuron in the previous layer, denoted by $w_i$.
> To adapt to the used notations of coresets, we define a function/mapping called $w:P\to R$, that maps every vector (neuron) $p_i$ to its weight (in the next layer) $w_i$, i.e., $w(p_i):= w_i$. This was done only to adapt to the coreset setting, where we are usually given a set of points $P$, and a weights function $w: P\to R$. It is **not** some kind of algebraic, polynomial, or any other function of the values, it is just a mapping from $p_i$ to $w_i$ to simplify the reading, writing, adaptations, citations, and usages of other theorems.
> Following your comment, we have clarified this in Line 83, by explaining that “ this is just a mapping from $p_i$ to $w_i$ to simplify the writing and reading.”
>
> ----
>
> **Q3:** What is the reason for assuming all the weights of P as equal to 1 in the main paper?
>
> **Answer:** As explained in Lines 103 -- 106 (of the original submitted version), to simplify the writing and reading, in the main paper, we gave the reader a coreset construction for the simpler case where the weights are all 1. In the appendix, our results are easily extended toward the case of having general weights; see the paragraph on weighted input in Section 2.5 (Extensions), and Section D in the appendix.
>
> ----
>
> **Q4:** In Section 2.1, between lines 114 and 116, what do you mean by "Once we bound these sensitivities" and "The size of the sample is proportional to the sum of these bounds"?
>
> **Answer:** Given a set of point $P$ in $R^d$, we wish to compute a coreset for these points via the sensitivity sampling framework. First, we need to bound the sensitivity of each point $p\in P$. The sensitivity of a point $p\in P$ is defined as $s(p) = \sup_{x\in X}\frac{\phi(p,x)}{\sum_{q\in P}\phi(q,x)}$ where the denominator is not zero.
>
> Hence, for every $p\in P$, we wish to compute a number $s’(p)$, such that $s’(p)\geq s(p)$. Once the bound $s'(p)$ on the sensitivity $s(p)$ of each point $p$ is computed, we can define $T=\sum_{p\in P} s’(p)$ as the total sensitivity (the sum of these bounds). Now, to obtain a coreset, we can sample points according to the distribution $s’(p)/T$, i.e., we sample $m>0$ points from $P$, where at each sample, the point $p\in P$ is sampled i.i.d with probability $s’(p)/T$.  We also re-weight the sampled points to obtain a coreset.
> As the bound $s’(p)$ (on $s(p)$) is tighter, the total sensitivity $T$ gets smaller, and then the coreset size $m$ (required number of sampled points) gets smaller, and vice versa.
> This is given formally in Theorem B.2, and we refer to this theorem in the text at the end of the paragraph “Computing pruning coresets via sensitivity sampling”.
>
> We have added this explanation to the appendix for the interested reader following your comment; see Subsection “Sensitivity Sampling Missing Details”, we also refer to it in the main paper.
>
> ----

---

> ### Author Response · Authors · 2022-08-02
> **Response to Reviewer 8WWu: Part 3. Answering questions 5 - 12 raised by the reviewer**
>
> We address the remaining questions.
>
> **Q5:** In Theorem 2.3, line 127, what do you mean by "points with nonempty interior"? Points do not have interior.
>
> **Answer:** What is meant by this is that the convex hull of these points has a nonempty interior. We thank the reviewer for pointing this out. We rephrased the theorem as follows:
> “Let $L\subseteq R^d$ be a set of points such that the convex hull of $L$ has a nonempty interior. Then, there exists an ellipsoid …”
>
> ----
>
> **Q6:** In Section 2.2, line 133, can you explain what do you mean by the dilated form of B consisting of a conical combination? Wouldn't it make more sense to say that this is a weighted sum in which the sum of the weights now exceeds 1?
>
> **Answer:** The dilated form of $B$ is formulated as the following set: $\left\lbrace \alpha (x - c) + c \mid x \in B \right\rbrace$, where $c$ here denotes the center of $B$. The right term in the field of Linear algebra for weighted sum where the weights sum up to a scalar larger than $1$ is denoted by canonical combination. Following this comment, we further clarified this directly after the first occurrence of the term "dilated form".
>
> ----
>
> **Q7:** In Section 3, why is every network pruned with a different subset of approaches? What happened with the missing ones in each of those cases?
>
> **Answer:** The reviewer raises a good point. Unfortunately, not all competing methods did take into account all the networks and datasets which we have tested in our paper. Thus, as common in the literature on network pruning, we collected the relevant methods for each tested network (for each tested network, we report the methods that have conducted experiments on it).
>
> ----
>
> **Q8:** In Table 1, what is the meaning of "Baseline Err." and "Pruned Err."? Is the first the complement of the accuracy and the second the drop due to pruning? Why is the first one almost the same but not always across all methods tested?
>
> **Answer:** The "Baseline Err" is the error of classification (percentage of misclassified examples from the test set) of the uncompressed model. The "Pruned Err" is the classification error of the compressed model. We have further clarified that in the paragraph "The setting" following this comment.
>
> ----
>
> **Q9:** In Table 2, what is the meaning of "FR"?
>
> **Answer:** The "FR" stands for (percentage of) floating point reduction. We refer the reviewer to Lines 244-249 of the original submitted version.
>
> ----
>
> **Q10:** In Table 3, is "Orig. Err." the same as "Baseline Err." from before? If so, please make it uniform.
>
> **Answer:** We thank the reviewer for this suggestion. It has been incorporated.
>
> ----
>
>
> **Q11:** Why is only one pruning rate reported for each model and approach?
>
> **Answer:** First, we note that we are working now on adding more tests on different compression rates. However, we must note two things:
>
> - (i) In all of the tested networks, we tried to report the **highest** compression rate across all methods, or at least to be very close to it, while achieving better or comparable results with all baselines. Hence, we think that having a higher compression rate and the best accuracy across all networks, shows the robustness of our approach.
>
> - (ii) In the context of pruning it is common to compare results by tables, where each reported method has one compression rate.
>
> ----
>
> **Q12:** How do you think coreset-based pruning relates to what happens in the case of exact compression methods?
>
> **Answer:** Unlike exact compression methods, coresets (usually) lead to compression with approximation guarantees. Theoretically speaking, coresets guarantee that for every query (not only for the optimum), its cost with respect to the coreset approximates the cost of the same query on the whole data. We believe that this theoretical guarantee in the context of exact coresets (i.e., coresets which provide zero approximation error for all queries) is hard to obtain for problems that are mathematically hard (such as model compression).
>
> We must note two things: (i) recent progress on exact coresets (such as "Fast and Accurate Least-Mean-Squares Solvers for High Dimensional Data"), showed that for some (probably easier) problems, exact coreset can be suggested, these problems include linear/lasso/ridge regression and PCA.  (ii) Practically, we can see that for some compression rates the coreset gives 0 error in network pruning or even improves the accuracy.

---

> > ### Comment · Reviewer_8WWu · 2022-08-07
> > **Following up**
> >
> > I appreciate the effort of the authors to improve the writing of the paper.
> >
> > I am still not satisfied with the limited computational evaluation presented the paper and I do not think that a similar practice in other published papers is a valid justification, but I am trusting that the authors will honor their promise to extend their evaluation further.

---

> > > ### Author Response · Authors · 2022-08-08
> > > **Additional experiments**
> > >
> > > We are grateful for the reviewer's appreciation of our efforts. As promised, we have extended our experiments to account for different compression rates. Specifically, for Vgg16 (on CIFAR-10), Vgg19 (on CIFAR-10), and Lenet (on MNIST), showing that regardless of the high compression rates, our coreset pruning mechanism results in favorable small models with high accuracy rates.
> > >
> > > We also note, that more experiments on Resnet-50 (for ImageNet) are being conducted. Indeed it will be ready by the time the camera-ready version is available.
> > >
> > > If you have any concerns please let us know.

---

> ### Author Response · Authors · 2022-08-02
> **Response to Reviewer 8WWu: Part 4.  Addressing other comments**
>
> We now address the general comments given by the reviewer:
>
> **Comment 1:** Some concepts are used before definition, some explanations are repeated, and often it is only possible to understand what is going on when a new and more accessible definition is presented.
>
> **Answer:** We fixed as much as we can following this comment and the detailed comments section provided by the reviewer. Furthermore, following your feedback, the entire paper had considerable writing revisions, not limited to the parts you commented on.
> Indeed, We will be happy to incorporate more and fix any other instances of such comments you have.
>
> ----
>
> **Comment 2:** what really concerns me is the limited experimental evaluation: the last sentence of the abstract sounds an alarm of hand-picking. Indeed, the results in Section 3 are very limited: only one pruning rate is used for each method tested, which is far from ideal. I would not expect a new method to always beat all other methods in order to accept a paper, but I do expect to see a thorough assessment of where and why a new method performs better.
>
> **Answer:** First, we note that we are working now on adding more tests on different compression rates. However, we note three things:
>
> - (i) In all of the tested networks, we tried to report the highest compression rate across all methods, or at least to be very close to it, while achieving better or comparable results with all baselines. Hence, we think that having a higher compression rate and the best accuracy (or very close to it) across all networks shows the robustness of our approach.
>
> - (ii) In the context of pruning it is common to compare results by tables, where each reported method has one compression rate.
>
> - (iii) The last sentence in the abstract is definitely not hand-picking. We wrote it in the abstracts since it is a very strong practical result in the context of coreset-based pruning. Specifically, since it was achieved on the ImageNet dataset. In Table 3, one can see that we have the highest compression rate and a comparable drop in the accuracy with methods that compressed much less. However, if you suggest removing it, we don't mind.
>
> ----
>
> **Comment 3**: they also need to provide more context for the readers by going over their design choices in more detail and with more context and explanations to non-experts.
>
> **Answer:** The reviewer raised a fruitful comment to clarify our contributions. In a nutshell, our coreset is generic in the sense that it can be used as is for different $l_p$ regression losses for any $p$ between $0$ and $\infty$. Such property is scarce in the literature on coresets and in our paper, we exploited such trait for the task of model compression.
>
> We modified the current manuscript to highlight this and clarify our choice of design further so that it will be understood by all.
>
> ----------
>
> **Again, thank you for the clear and detailed comments.**

---

### Author Response · Authors · 2022-08-04
**Summary of Reviews and Discussions: Concerns Are Addressed**

Dear Reviewers, ACs, and SACs,

First, we would like to express our deep gratitude and appreciation for spending the time to review and assess our paper and for providing us with insightful comments. Your feedback has already helped us to improve our paper significantly.

We are happy to update you that we have uploaded a new (improved) version of our manuscript following your insightful review.  Overall, we believe that in our response and the new version were able to address and answer the reviewers’ concerns to the extent that will hopefully convince the reviewers to raise their scores.

We look forward to further engaging with you during the upcoming discussion period.

Thank you.

---

### Author Response · Authors · 2022-08-08
**Thank You**

Dear Reviewers, ACs, and SACs,

We would like to emphasize our appreciation for the openly communicated review process, and to the reviewers who made this process very valuable and beneficial by communicating with us, asking and answering questions, suggesting many helpful suggestions, and raising their scores following the fruitful discussion - we truly appreciate that.

As the discussion period is coming to an end, we would like to note that we are here to address any lingering issues and comments at short notice. At the same time, we believe we were able to more than adequately address all the reviewers’ concerns and issues.

Thank you and we look forward to hearing back,
The authors

---

### Meta-Review · Area_Chair_zatd · 2022-08-23

**Recommendation:** Accept
**Confidence:** Less certain

**Metareview:**

The paper proposes a method for pruning neural network weights at each layer. The authors have addressed the concerns from the reviewers and the reviewers raised their scores. Although the reviewer somewhat agrees with the authors that the standard practice in the network pruning literature is to report a very limited set of results, the reviewer expects the authors to try a bit harder to expand the results for the final revision. Please incorporate the reviewers’ suggestions in their detailed reviews and revise the final version of the paper properly.

**Award:**

No

---

### Decision · Program_Chairs · 2022-09-14

Accept